# PRMT1-mediated methylation of MICU1 determines the UCP2/3 dependency of mitochondrial $Ca^{2+}$ uptake in immortalized cells

Corina T. Madreiter-Sokolowski[1], Christiane Klec[1], Warisara Parichatikanond[1,†], Sarah Stryeck[1], Benjamin Gottschalk[1], Sergio Pulido[2], Rene Rost[1], Emrah Eroglu[1], Nicole A. Hofmann[1], Alexander I. Bondarenko[1], Tobias Madl[1,3,4], Markus Waldeck-Weiermair[1], Roland Malli[1] & Wolfgang F. Graier[1]

Recent studies revealed that mitochondrial $Ca^{2+}$ channels, which control energy flow, cell signalling and death, are macromolecular complexes that basically consist of the pore-forming mitochondrial $Ca^{2+}$ uniporter (MCU) protein, the essential MCU regulator (EMRE), and the mitochondrial $Ca^{2+}$ uptake 1 (MICU1). MICU1 is a regulatory subunit that shields mitochondria from $Ca^{2+}$ overload. Before the identification of these core elements, the novel uncoupling proteins 2 and 3 (UCP2/3) have been shown to be fundamental for mitochondrial $Ca^{2+}$ uptake. Here we clarify the molecular mechanism that determines the UCP2/3 dependency of mitochondrial $Ca^{2+}$ uptake. Our data demonstrate that mitochondrial $Ca^{2+}$ uptake is controlled by protein arginine methyl transferase 1 (PRMT1) that asymmetrically methylates MICU1, resulting in decreased $Ca^{2+}$ sensitivity. UCP2/3 normalize $Ca^{2+}$ sensitivity of methylated MICU1 and, thus, re-establish mitochondrial $Ca^{2+}$ uptake activity. These data provide novel insights in the complex regulation of the mitochondrial $Ca^{2+}$ uniporter by PRMT1 and UCP2/3.

[1] Center for Molecular Medicine, Institute of Molecular Biology and Biochemistry, Medical University of Graz, Harrachgasse 21/III, Graz 8010, Austria. [2] Institute of Chemistry, University of Graz, Graz 8010, Austria. [3] Center for Integrated Protein Science, Department Chemistry, Technical University Munich, Garching 85748, Germany. [4] Institute of Structural Biology, Helmholtz Zentrum München, Neuherberg 85764, Germany. † Present address: Department of Pharmacology, Mahidol University, Bangkok 10400, Thailand. Correspondence and requests for materials should be addressed to W.F.G. (email: wolfgang.graier@medunigraz.at).

The identity of mitochondrial $Ca^{2+}$ channels that achieve mitochondrial $Ca^{2+}$ uptake has been elusive for a long time. By identifying mitochondrial $Ca^{2+}$ uptake 1 (MICU1) as an essential component of mitochondrial $Ca^{2+}$ uniport, Mootha and colleagues[1] achieved the major breakthrough in solving the long-awaited molecular identities of proteins that actually accomplish mitochondrial $Ca^{2+}$ uptake[1]. Based on this landmark study, other key components of the mitochondrial $Ca^{2+}$ uptake machinery including the mitochondrial uniporter MCU[2,3], its dominant-negative form MCUb[4], MICU2 (ref. 5) and the essential MCU regulator (EMRE)[6] of which the topology remains controversial[7–10], have been identified, emphasizing that mitochondrial $Ca^{2+}$ uptake is accomplished by a sophisticated heteromeric protein complex[11–14]. Importantly, MICU1 was found to act as gatekeeper for the MCU complex[15,16]. Notably, additional proteins including the mitochondrial calcium uniporter regulator 1 (MCUR1)[10,17,18], solute carrier family 25A23 (SLC25A23)[19], leucine zipper EF-hand containing transmembrane protein 1 (Letm1)[20,21], and uncoupling proteins 2 and 3 (UCP2/3)[22] have been reported to contribute to mitochondrial $Ca^{2+}$ uptake. However, the molecular mechanisms how these proteins actually add to or influence MCU activity remain largely unresolved and the direct regulation of mitochondrial $Ca^{2+}$ uptake by these proteins has been questioned recently[14,23,24]. Accordingly, ongoing research aims to shed light on the complex function of mitochondrial $Ca^{2+}$ uptake machineries in different cell types. Notably, UCP2/3 were found to be fundamental for efficient mitochondrial $Ca^{2+}$ uptake in all our related studies[22,25,26] and that of others[27–30]. Moreover, UCP2 was described to selectively regulate a MCU-dependent extra-large mitochondrial $Ca^{2+}$ current in mitoplasts isolated from HeLa cells[31] and to contribute to mCa1 current in mitoplasts isolated from cardiomyocytes[32]. Nevertheless, in other reports no evidence for an engagement of UCP2/3 in mitochondrial $Ca^{2+}$ signalling was found[20,33,34]. Accordingly, the present work seeks to solve controversy on the engagement of UCP2/3 in mitochondrial $Ca^{2+}$ uptake.

Here we show that posttranslational protein modification of MICU1 regulates the activity of the mitochondrial $Ca^{2+}$ uniporter complex. In particular, protein arginine methyl transferase 1 (PRMT1)-mediated methylation of MICU1 desensitizes this protein for $Ca^{2+}$ and, in turn, reduces mitochondrial $Ca^{2+}$ uptake. UCP2 binds exclusively to the methylated form of MICU1, normalizes its $Ca^{2+}$ sensitivity and re-establishes mitochondrial $Ca^{2+}$ uptake. Thus, UCP2/3 are described as sensitizer of mitochondrial $Ca^{2+}$ uptake under conditions of increased PRMT1 activity, a phenomenon that has probably great importance in cancer cells.

## Results

**Diverse contribution of UCP2 to mitochondrial $Ca^{2+}$ uptake.** First, we searched for cell types with and without an obvious role of UCP2/3 in mitochondrial $Ca^{2+}$ uptake. As previously reported, small interfering RNA (siRNA)-mediated knockdown of UCP2/3 yielded strong reduction of mitochondrial $Ca^{2+}$ uptake on intracellular $Ca^{2+}$ release by any $IP_3$-generating agonist in HeLa cells (Fig. 1a) and Ea.hy926 cells (Fig. 1b), a hybridoma cell line that was established by fusing primary human umbilical vein endothelial cells (HUVECs) with a thioguanine-resistant clone of human lung carcinoma (A549)[35]. In contrast, in freshly isolated, short-term cultured HUVECs and primary porcine aortic endothelial cells (PAECs) knockdown of UCP2/3 was without an obvious effect on mitochondrial $Ca^{2+}$ uptake under the same experimental conditions (Fig. 1c,d), although knockdown

efficiency determined by reverse transcriptase–PCR (RT–PCR) was comparable in all four cell lines (Supplementary Fig. 1). However, knockdown of MCU yielded a strong reduction of mitochondrial $Ca^{2+}$ uptake on intracellular $Ca^{2+}$ release in all four cell types (Supplementary Fig. 2), confirming that MCU is a ubiquitous core protein of the mitochondrial $Ca^{2+}$ uniporter complex. Notably, the effect of UCP2/3 knockdown on mitochondrial $Ca^{2+}$ sequestration in HeLa and Ea.hy926 cells was independent from mitochondrial membrane potential, sarco/endoplasmic reticulum (ER) $Ca^{2+}$ ATPase activity, and activity of mitochondrial $Na^+/Ca^{2+}$ exchanger[22], pointing to a direct regulation of the MCU activity by UCP2/3 in these cell types.

Considering the close relationship between the immortalized HUVEC cell line Ea.hy926 and primary HUVECs, the opposite sensitivity of mitochondrial $Ca^{2+}$ uptake to UCP2/3 knockdown in these cell types of same origin was surprising and offered us the possibility to further explore the principles that are responsible for an engagement of UCP2/3 in mitochondrial $Ca^{2+}$ uptake. In view of the various and inconclusive messenger RNA expression patterns of some constituents of the MCU complex in all cell types used (Supplementary Fig. 3), we speculated that differences in the posttranslational protein modification(s) of one or more core proteins of the mitochondrial uniporter complex might cause the UCP2/3 dependency or independency of mitochondrial $Ca^{2+}$ uptake in the various cell types.

**PRMT1 activity introduces UCP2/3 as regulators of MCU.** As in recent studies the protein arginine methylation via PRMT1 was found to regulate multiple protein functions[36–38] and mitochondria-associated cell signalling pathways[39], we investigated whether protein arginine methylation is involved in the regulation of the mitochondrial $Ca^{2+}$ uniporter activity and determines the UCP2/3 dependency of mitochondrial $Ca^{2+}$ uptake. In all cells tested, several PRMT isoforms could be detected by PCR (Supplementary Fig. 4). As PRMT1 is the main protein for asymmetric protein arginine methylation[40,41], the overall methylation status in Ea.hy926 cells was compared with that of freshly isolated and short-term cultured HUVECs using a specific antibody against asymmetric protein arginine methylation (aDMA). Notably, the overall asymmetric protein arginine methylation was much stronger in the Ea.hy926 cells than in native HUVECs (Fig. 2a,b). This finding unveiled that the degree of asymmetric protein arginine methylation positively correlates with the UCP2/3 sensitivity of mitochondrial $Ca^{2+}$ uptake in these closely related cell types. In line with these findings, HeLa cells that exhibit UCP2/3 sensitivity of mitochondrial $Ca^{2+}$ uptake had a very strong level of asymmetric protein arginine methylation, whereas PAECs that did not show UCP2/3-sensitivity of mitochondrial $Ca^{2+}$ uptake were found to contain similar low PRMT1 activity as native HUVECs (Fig. 2a,b). These correlations suggest that a strong PRMT1 activity engages UCP2/3 to facilitate mitochondrial $Ca^{2+}$ uptake. To evaluate this hypothesis, we incubated HeLa cells, that exhibit an UCP2/3-sensitive mitochondrial $Ca^{2+}$ uptake, with the methylation inhibitor Adox[42] and the selective PRMT1 inhibitor AMI-1 (ref. 43 and Supplementary Fig. 5) for 72 h and tested whether an inhibition of the PRMT1 activity affects the UCP2/3 sensitivity of mitochondrial $Ca^{2+}$ uptake. The pharmacological inhibition of PRMTs/PRMT1 indeed abolished the UCP2/3 dependency of mitochondrial $Ca^{2+}$ uptake in HeLa cells, while no other change in mitochondrial $Ca^{2+}$ signals became obvious (Fig. 2c). In line with these findings that point to an important role of PRMT1 in the engagement of UCP2/3 in mitochondrial $Ca^{2+}$ uptake, knockdown of PRMT1 by specific

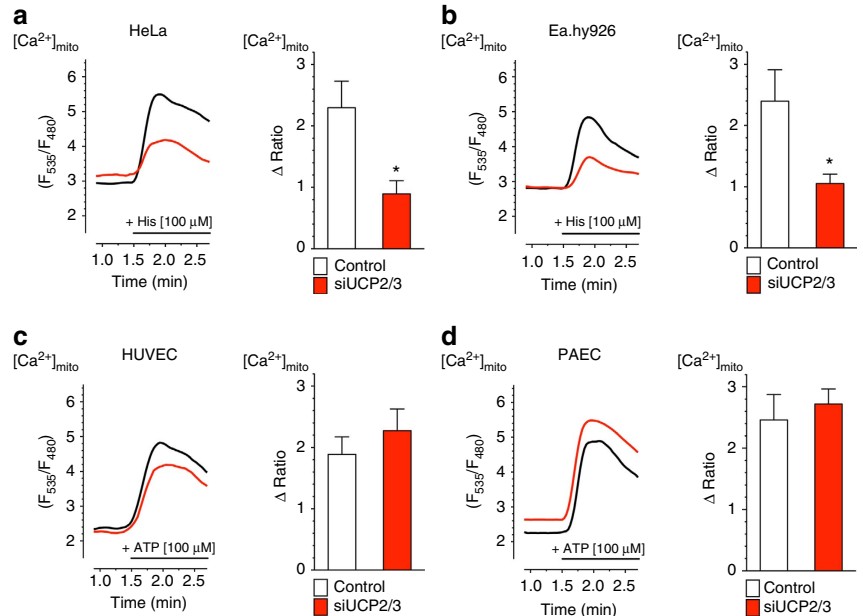

**Figure 1 | Cell type specific differences in the UCP2/3 dependency of mitochondrial Ca$^{2+}$ uptake.** Left panels: representative curves reflect mtCa$^{2+}$ ratio signals over time of (**a**) HeLa, (**b**) Ea.hy926, (**c**) HUVEC and (**d**) PAEC cells expressing 4mtD3cpv in response to agonists measured in Ca$^{2+}$-free solution in control cells (black curves) or cells with knockdown of UCP2/3 (red curves). To induce Ca$^{2+}$ release from intracellular ER, 100 μM histamine (His) was applied in HeLa and Ea.hy926 cells, whereas 100 μM ATP was used as an agonist in HUVEC and PAEC cells. Right panels: bars represent an average of maximal Δ ratio signals (mean ± s.e.m.) on cell treatment with IP$_3$ generating agonists in Ca$^{2+}$-free solution of control cells (white columns; *HeLa*: n = 35/13, *Ea.hy926*: n = 29/13, *HUVEC*: n = 37/15 and *PAEC*: n = 17/10) and cells treated with siRNA against UCP2/3 (red columns; *HeLa*: n = 26/12, *Ea.hy926*: n = 30/13, *HUVEC*: n = 26/12 and *PAEC*: n = 18/10). Numbers indicate the numbers of cells per independent repeats. *P < 0.05 versus control using the unpaired Student's *t*-test.

siRNAs abolished the UCP2/3 sensitivity of mitochondrial Ca$^{2+}$ uptake in Ea.hy926 (Fig. 2d) and HeLa cells (Fig. 2e). In contrast, siRNA-mediated reduction of PRMT2, PRMT3, PRMT4 and PRMT6 had no effect on the UCP2/3 dependence of mitochondrial Ca$^{2+}$ uptake in HeLa cells (Supplementary Fig. 6).

Next, PRMT1 was transiently overexpressed in the native HUVECs, the cell type exhibiting an UCP2/3-independent mitochondrial Ca$^{2+}$ uniport. Overexpression of PRMT1 in HUVECs yielded introduction of UCP2/3 dependency of mitochondrial Ca$^{2+}$ uptake on intracellular Ca$^{2+}$ release by an IP$_3$-generating agonist, whereas no other change in the mitochondrial Ca$^{2+}$ signalling was observed (Fig. 2f). Neither an overexpression of PRMT1 nor its knockdown altered cytosolic Ca$^{2+}$ signals (Supplementary Fig. 7), the ER Ca$^{2+}$ content (Supplementary Fig. 8), basal mitochondrial Ca$^{2+}$ concentration in HeLa, Ea.hy926 or HUVECs (Supplementary Fig. 9), mRNA expression levels of MCU, MCUb, EMRE, MICU1, MCUR1 or UCP2 (Supplementary Fig. 10), and the physical interaction of the ER with mitochondria (Supplementary Fig. 11). Only the expression of UCP3 was altered in cells with knockdown or overexpression of PRMT1 (Supplementary Fig. 10). As the expression of UCP2 is about 250-fold higher than the expression level of UCP3 (Supplementary Fig. 10), it is rather unlikely to be that these changes have an impact on mitochondrial Ca$^{2+}$ uniport. Thus, these data indicate that PRMT1-mediated protein methylation directly engages UCP2/3 as fundamental regulators of the MCU activity.

**UCP2 normalizes Ca$^{2+}$ uptake upon PRMT1 activity.** To understand how UCP2/3 contribute to mitochondrial Ca$^{2+}$ uptake in a PRMT1-dependent manner, we next explored the interrelation between PRMT1-mediated protein arginine methylation and UCP2/3 by evaluating the Ca$^{2+}$ sensitivity of mitochondrial Ca$^{2+}$ uptake in permeabilized HeLa cells that were depleted by respective siRNA(s) from UCP2/3 and/or PRMT1. The Ca$^{2+}$ sensitivity of mitochondrial Ca$^{2+}$ uptake decreased in PRMT1-active wild-type HeLa cells from 5.3 (4.1–6.7) μM in controls to 14.1 (12.0–16.5) μM in cells that were depleted from UCP2/3 (Fig. 3a), indicating that UCP2/3 facilitate mitochondrial Ca$^{2+}$ uniport by enhancing the Ca$^{2+}$ sensitivity of the mitochondrial Ca$^{2+}$ uptake machinery. In contrast, HeLa cells treated with siRNA against PRMT1 did not show any UCP2/3 dependency of the Ca$^{2+}$ sensitivity of mitochondrial Ca$^{2+}$ sequestration (with UCP2/3: EC$_{50}$ = 5.3 (4.1–6.8) μM Ca$^{2+}$; without UCP2/3: EC$_{50}$ = 4.7 (3.5–6.3) μM Ca$^{2+}$; Fig. 3b). These data indicate that in the presence of elevated PRMT1 activity, UCP2/3 serve as sensitizers of the mitochondrial Ca$^{2+}$ uptake machinery.

**MICU1 is the target for PRMT1 within the MCU complex.** The activity of the mitochondrial Ca$^{2+}$ uniporter complex to allow Ca$^{2+}$ influx into the mitochondrial matrix essentially requires MCU[2,3] and EMRE[6], and is tightly controlled by the binding of Ca$^{2+}$ to the EF-hand domains at the carboxy-terminal end of MICU1 (refs 1,16,44). Computational sequence analyses revealed several potential arginine methylation sites in MICU1, MCU, UCP2 and EMRE (Supplementary Table 1)[45]. As the predicted methylation site of EMRE is in or very close to the predicted transmembrane domain[6], only MCU, MICU1, and UCP2 were further tested as putative targets of PRMT1 using co-immuno-precipitation of either of these proteins with antibodies specifically recognizing mono- and dimethyl, as well as asymmetric dimethyl (aDMA) protein arginine methylation, respectively. While neither MCU (Fig. 3c) nor UCP2 (Fig. 3d) were methylated by PRMT1, MICU1 got strongly methylated by this particular methyl transferase (Fig. 3e and Supplementary

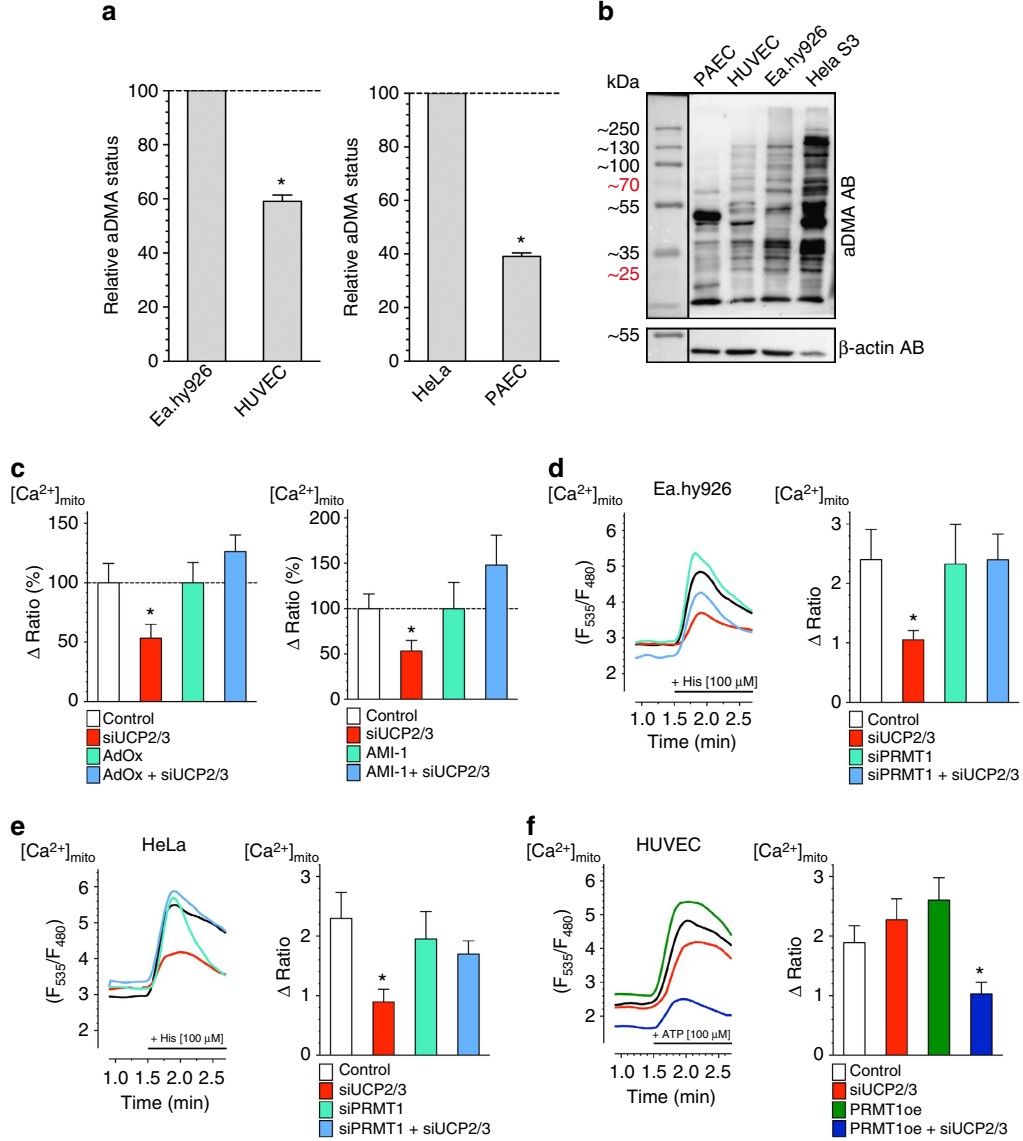

**Figure 2 | Methylation by PRMT1 engages UCP2/3 in mitochondrial Ca$^{2+}$ uptake.** (**a**) Percentage of asymmetric arginine dimethlyation (aDMA) in HUVECs normalized to aDMA state of Ea.hy926 ($n = 3$), as well as percentage of aDMA of PAECs normalized to dimethlyation state of HeLa ($n = 4$). (**b**) Representative western blots revealing the overall methylation state and housekeeping protein β-actin level of HeLa, Ea.hy926, HUVEC and PAEC cells by using an antibody against asymmetric dimethyl arginine motif aDMA and β-actin, respectively. (**c**) Bars show maximal Δ ratio values in response to 100 μM histamine in Ca$^{2+}$-free conditions of 4mtD3cpv-expressing HeLa cells treated with control siRNA (white columns: Control: $n = 82/15$), siRNA against UCP2/3 (red columns; *siUCP2/3*: $n = 80/15$), a 72 h incubation of PRMT1 inhibitors, either 40 μM AdOx (left panel) or 5 μM AMI-1 (right panel) (light green columns; *AdOx*: $n = 33/7$, *AMI-1*: $n = 66/8$), or a combination of UCP2/3 knockdown and methylation inhibitors (light blue columns; *AdOx*: $n = 37/8$, *AMI-1*: $n = 92/25$). Data were calculated as percentage of maximal Δ ratio values of HeLa cells with control siRNA (mean ± s.e.m.). Left panels: representative curves indicate mtCa$^{2+}$ ratio signals over time of (**d**) Ea.hy926 and (**e**) HeLa cells expressing 4mtD3cpv on stimulation with 100 μM histamine (His) under Ca$^{2+}$-free conditions of control (black curves), siRNA against UCP2/3 (red curves) or PRMT1 (light green curves), or a combination of both (light blue curves). Right panels: bars represent an average of maximal Δ ratio signals after exposure with an IP$_3$ generating agonist in cells treated with control siRNA (white columns; *Ea.hy926*: $n = 29/13$, *HeLa*: $n = 35/13$), siUCP2/3 (red columns; *Ea.hy926*: $n = 30/13$, *HeLa*: $n = 26/12$), siPRMT1 (light green columns; *Ea.hy926*: $n = 31/12$, *HeLa*: $n = 26/9$) or an combination of siUCP2/3 and siPRMT1 (light blue columns; *Ea.hy926*: $n = 24/13$, *HeLa*: $n = 32/13$). (**f**) Left panel: representative curves show mtCa$^{2+}$ ratio signals over time in response to 100 μM ATP in Ca$^{2+}$-free solution in HUVEC cells with control siRNA (dark curve), siRNA against UCP2/3 (red curve), overexpression of PRMT1 (dark green curve) or a combination of both (dark blue curve). Right panel: bars represent an average of maximal Δ ratio signals of IP$_3$ agonist stimulated response of control cells (white column; $n = 37/15$), cells overexpressing PRMT1 (dark green column; $n = 30/12$), cells depleted of UCP2/3 (red column; $n = 26/12$) or a combination of both (dark blue column; $n = 38/13$). Numbers indicate the numbers of cells/independent repeats. *$P < 0.05$ versus control using the unpaired Student's t-test.

Fig. 12). Purity of immunoprecipitation procedure was verified by comparing IgG control versus control cell lysates (Supplementary Fig. 13). These data point to MICU1 as potential target of PRMT1.

**Methylation at position 455 of MICU1 engages UCP2.** Analyses of the 13 potential arginine methylation sites in the MICU1 protein regarding their sequence conservation, structural accessibility, and localization in regulatory regions revealed arginine

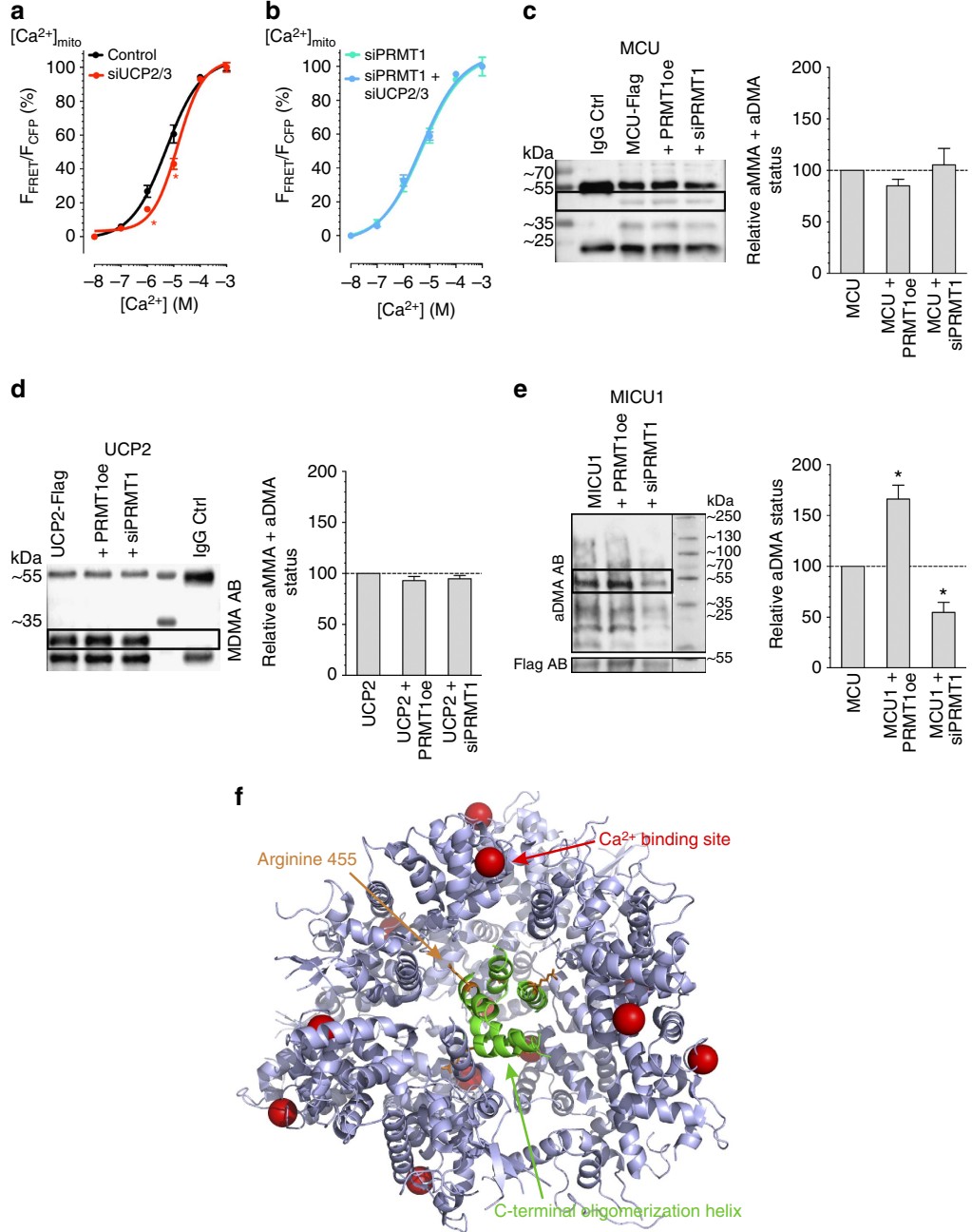

**Figure 3 | MICU1 is the methylation target of PRMT1 within the mitochondrial Ca$^{2+}$ uptake machinery.** (**a**) Concentration response curves show mtCa$^{2+}$ uptake in HeLa cells with (red curve; $n = 42/7$) or without (black curve; $n = 16/4$) depletion of UCP2/3 treated with 3 μM of ionomycin and as indicated with different concentrations of Ca$^{2+}$. (**b**) Concentration response curves of mitochondrial Ca$^{2+}$ uptake in HeLa cells with depletion of PRMT1 (light green curve; $n = 28/6$) or combined depletion of PRMT1 and UCP2/3 (light blue curve; $n = 24/6$) treated with 3 μM ionomycin and as indicated with different concentrations of Ca$^{2+}$. Left panels: representative western blots revealing arginine methylation of (**c**) MCU, (**d**) UCP2 or (**e**) MICU1, after immunoprecipitation of flag-tagged proteins, by using an antibody against mono- and dimethyl arginines (MMA/SDMA) or asymmetric dimethyl arginines (aDMA) and normalized to the intensities of detected MCU, UCP2, MICU1, or Flag bands, respectively. For **c** and **d**, MDMA (mono- and dimethyl) antibody was used and bands were normalized to MCU or UCP2 content, respectively. For **e**, aDMA antibody was used and bands were normalized to the intensity of the Flag bands. Right panels: bars represent the percentage of arginine methylation of flag-tagged proteins of cells with knockdown or overexpression of PRMT1 calculated as percentage of (**c**,**d**) mono- and dimethyl or (**e**) asymmetric dimethyl arginine methylation of flag-tagged proteins of control cells. (**f**) Crystal structure of the MICU1 hexamer (PDB 4NSC)[47]. The C-terminal oligomerization helices, Ca$^{2+}$ binding sites, and arginine 455 are highlighted. *$P < 0.05$ versus control using the unpaired Student's $t$-test.

455 as the most promising candidate (Supplementary Table 2). This arginine residue is localized within the C-terminal region of MICU1, which extends into the intermembrane space (Fig. 3f). Based on this *in silico* approach, we cloned two MICU1 mutants in which the arginine at position 455 was exchanged by either lysine (MICU1-K) or phenylalanine (MICU1-F). Both MICU1 mutants could not be methylated by PRMT1 (Fig. 4a and Supplementary Fig. 14) and mimic either the non-methylated (that is MICU1-K) or methylated MICU1 (that is MICU1-F) wild type[46]. Both mutated MICU1 proteins localized at mitochondria

(Supplementary Fig. 15). Expression of neither MICU1-K nor MICU1-F in HeLa and PAEC cells affected basal mitochondrial $Ca^{2+}$ levels compared with cells overexpressing wild-type MICU1 (Supplementary Fig. 16). Both MICU1 mutants showed unchanged patterns of co-localization with MCU or EMRE (Supplementary Fig. 17). However, the mitochondrial morphology was slightly affected by the MICU1-F compared with MICU1 and MICU1-K (Supplementary Fig. 18). As neither the PRMT1 knockdown nor PRMT1 overexpression affected mitochondrial shape, we assume that these morphological changes are specific for MICU1-F mutant and not related to PRMT1-mediated methylation. To identify position 455 in MICU1 as the target for PRMT1 methylation, a long and a short version of MICU1 were generated ($MICU1_{97-476}$ and $MICU1_{97-444}$). Investigation of the asymmetric protein arginine methylation status with the aDMA antibody demonstrated that only the long version, containing arginine at position 455, could be methylated by PRMT1 (Supplementary Fig. 19). Next, we

compared the non-methylate-able MICU1-K with the wild-type MICU1 in HeLa cells that exhibit strong intrinsic PRMT1 activity (Fig. 2a). The non-methylated MICU1 mutants established an UCP2/3-independent mitochondrial $Ca^{2+}$ uptake (Fig. 4b). Subsequently, the methylation-mimicking mutant MICU1-F was compared with wild-type MICU1 in PAECs that showed a low intrinsic methylation level (Fig. 2a). Expression of MICU1-F in PAECs established UCP2/3 sensitivity of mitochondrial $Ca^{2+}$ uptake (Fig. 4c). These data emphasize that a PRMT1-mediated methylation of arginine at position 455 of MICU1 is essential for the engagement of UCP2/3 in mitochondrial $Ca^{2+}$ uptake.

**UCP2 binds to methylated MICU1 and normalizes $Ca^{2+}$ binding.** Analysis of the published crystal structure of MICU1 revealed that R455 is localized in the C terminus (Fig. 3f), a region that has been identified to stabilize MICU1 hexamers[47]. As R455 asymmetric dimethylation could modulate either MICU1 oligomerization, $Ca^{2+}$ binding, or both, a detailed analysis of these properties *in vitro* using the purified proteins was performed. As reported previously[47], MICU1 exists in several oligomeric species. *In vitro* arginine methylation using recombinant PRMT1 caused severe MICU1 precipitation that

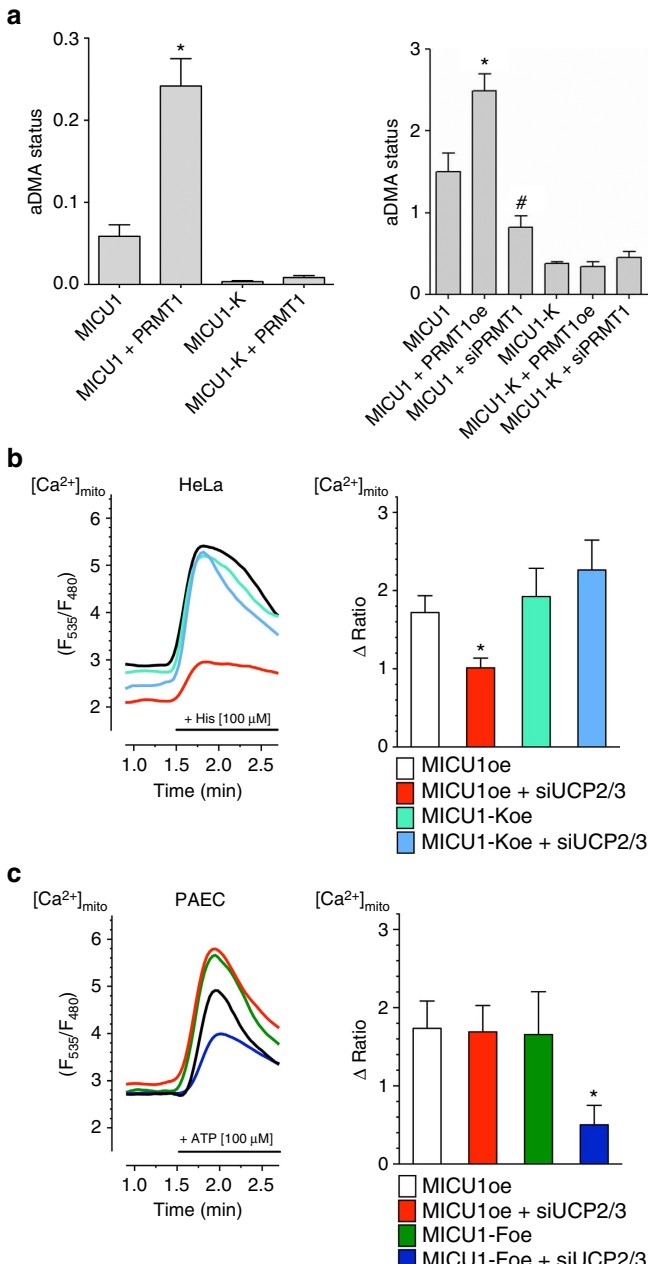

**Figure 4 | Methylation at position 455 of MICU1 is crucial for the engagement of UCP2/3 in mitochondrial $Ca^{2+}$ uptake.** (**a**) Left panel: bars show aDMA status of MICU1 constructs MICU1-WT and MICU1-K mutant with or without prior *in vitro* methylation by recombinant PRMT1, normalized to MICU1 content (*MICU-WT*: $n = 7$; *MICU1-WT + PRMT1*: $n = 6$; *MICU1-K*: $n = 8$; *MICU1-K + PRMT1*: $n = 8$). aDMA status was determined by using aDMA antibody and normalized to MICU1 content. The statistically significant different aDMA status of MICU1-WT and MICU1-WT + PRMT1 was marked with a star. Right panel: bars represent the aDMA status of flag-tagged MICU1-WT and MICU1-K proteins of control cells or cells with either overexpression or knockdown of PRMT1 ($n = 3$). After immunoprecipitation of flag-tagged proteins, aDMA antibody was used to reveal asymmetric arginine methylation and bands were normalized to the intensities of detected bands using an anti-Flag antibody. The statistically significant different aDMA status between MICU1-WT and MICU1-WT + PRMT1 was marked with a star and the statistically significant difference between MICU1-WT and MICU1 + siPRMT1 with a hash. (**b**) Left panel: representative curves show mtCa$^{2+}$ ratio signals, measured by 4mtD3cpv, over time in response to 100 μM histamine in $Ca^{2+}$-free solution in HeLa cells overexpressing MICU1-WT with (red curve) or without (black curve) siRNA against UCP2/3 or overexpression of MICU1-K mutant with (light blue curve) or without (light green curve) siRNA against UCP2/3. Right panel: bars represent an average of maximal Δ ratio signals (mean ± s.e.m.) of IP$_3$ generating agonist treatment of HeLa cells with overexpression of MICU1-WT (white column: $n = 68/16$), overexpression of MICU1-WT and knockdown of UCP2/3 (red column: $n = 56/18$), overexpression of MICU1-K (light green column: $n = 39/16$) and combined overexpression of MICU1-K and knockdown of UCP2/3 (light blue column: $n = 37/16$). (**c**) Left panel: representative curves reflect mtCa$^{2+}$ ratio signals, measured by 4mtD3cpv, over time in response to 100 μM ATP under $Ca^{2+}$-free conditions of PAEC cells overexpressing MICU1-WT treated with (red curve) or without siRNA against UCP2/3 (black curve) or PAEC cells overexpressing MICU1-F mutant with (dark blue curve) or without (dark green curve) UCP2/3. Right panel: bars represent an average maximal Δ ratio signals (mean ± s.e.m.) of IP$_3$ generating agonist treatment of PAEC cells overexpressing MICU1-WT (white column: $n = 10/8$), overexpressing MICU1-WT and treated with siRNA against UCP2/3 (red column: $n = 8/7$), overexpressing MICU1-F (dark green column: $n = 13/6$) or overexpression of MICU1-F in combination with knockdown of UCP2/3 (dark blue column: $n = 15/9$). In the imaging experiments, numbers indicate the numbers of cells/independent repeats. *$P < 0.05$ versus control using the unpaired Student's *t*-test.

**Table 1 | MICU1 *in vitro* binding data.**

| Sample | $K_d$ (μM) |
|---|---|
| MICU1[97-476] + CaCl$_2$ | 5.4 ± 0.7 |
| meMICU1[97-476] + CaCl$_2$ | 14.0 ± 1.5 |
| MICU1[97-476] R455K + CaCl$_2$ | 6.8 ± 0.7 |
| (me)MICU1[97-476] R455K + CaCl$_2$ | 6.0 ± 0.9 |
| MICU1[97-476] + IML2[UCP2] | no binding |
| meMICU1[97-476] + IML2[UCP2] | 24.6 ± 17.7 |
| MICU1[97-476]/IML2[UCP2] + CaCl$_2$ | 14.7 ± 6.4 |
| meMICU1[97-476]/IML2[UCP2] + CaCl$_2$ | 17.1 ± 9.9 |

Dissociation constants ($K_d$) as determined by isothermal titration calorimetry. The errors shown correspond to the s.d. of curve fits obtained by the MicroCal Origin software version 7.0 (see Supplementary Fig. 5 for the ITC curves). The $K_d$ provide a measure of binding affinities of MICU1 for CaCl$_2$ and IML2[UCP2]. A low $K_d$ is characteristic for high affinity/strong binding, a high $K_d$ for low affinity/weak binding. $K_d$ were obtained by titrating either CaCl$_2$ or IML2[UCP2] solutions into a MICU1[97-476] or MICU1[97-476]/IML2[UCP2] protein solution. Arginine methylation increases the $K_d$ of MICU1 for CaCl$_2$ that is equivalent with weaker affinity. Binding of IML2[UCP2] to MICU1 restores the CaCl$_2$ binding affinity of arginine methylated MICU1. All ITC experiments in the absence of IML2[UCP2] were carried out in triplicates.

indicates formation of large aggregates. Nevertheless, sufficient protein amounts could be obtained and stabilized in arginine-containing buffer after methylation to study Ca$^{2+}$ binding *in vitro* using isothermal titration calorimetry (ITC). We found that methylation of MICU1 strongly reduced its sensitivity for Ca$^{2+}$ to trigger de-oligomerization from a dissociation constant ($K_d$) of 5.4 ± 0.7 μM for non-methylated MICU1 to 14.0 ± 1.5 μM binding of Ca$^{2+}$ to methylated MICU1 (Table 1 and Supplementary Fig. 20). This demonstrates that methylation of MICU1 reduces the protein's responsiveness to Ca$^{2+}$. Such a reduced Ca$^{2+}$ sensitivity of methylated MICU1 would considerably impair the activity of the mitochondrial Ca$^{2+}$ uniporter complex. ITC data of a MICU1-K single mutant protein showed no significant change in Ca$^{2+}$ binding before and after methylation reaction (6.8 ± 0.7 μM and 6.0 ± 0.9 μM, respectively), indicating that R455 is the only residue modified by PRMT1 (Table 1 and Supplementary Fig. 21). Notably, although non-methylated MICU1 exhibits adequate Ca$^{2+}$ sensitivity in the physiological range that have been reported within the ER–mitochondria junction (app. 4–16 μM)[48], methylation of MICU1 shifts its Ca$^{2+}$ sensitivity towards a 2.6-fold higher Ca$^{2+}$ level.

Hence, an exclusive role of UCP2/3 in the regulation of methylated MICU1 is postulated, thus explaining the engagement of UCP2/3 in cells with high but not with low or absent PRMT1 activity. To further investigate by which mechanism UCP2/3 might facilitate the Ca$^{2+}$-induced rearrangement of methylated MICU1, we tested for binding of UCP2 to methylated and non-methylated MICU1 using ITC. To avoid improper folding of the suspended membrane protein UCP2 in a liquid phase, we decided to use the chemically synthetized intermembrane loop 2 of UCP2 (IML2[UCP2]) that has been described to be essential for an engagement of UCP2/3 to mitochondrial Ca$^{2+}$ uptake[22,26]. Therefore, binding of IML2[UCP2] to purified non-methylated and methylated MICU1 and its impact on Ca$^{2+}$ sensitivity, were tested *in vitro* using ITC. These experiments unveiled that the IML2[UCP2] exclusively binds to methylated MICU1 with a $K_d$ of 24.6 ± 17.7 μM (Supplementary Fig. 22). In contrast, no binding of IML2[UCP2] to non-methylated MICU1 was detected (Supplementary Fig. 5c). Moreover, although IML2[UCP2] had no impact on the Ca$^{2+}$ sensitivity of non-methylated MICU1 (14.7 ± 6.4), it shifted the Ca$^{2+}$ sensitivity of methylated MICU1 (17.1 ± 9.9) towards that of the non-methylated MICU1 (5.4 ± 0.7; Table 1 and Supplementary Fig. 23). These data demonstrate that UCP2 via its IML2 interferes

preferentially with methylated MICU1. This interaction consequently facilitates the Ca$^{2+}$-mediated rearrangement of MICU1 aggregates.

**Methylation desensitizes MICU1 for Ca$^{2+}$-induced reordering**. To evaluate the impact of MICU1 methylation on the rearrangement of these proteins on elevating Ca$^{2+}$ concentrations in cells, we next applied Förster resonance energy transfer (FRET) imaging in HeLa cells co-expressing MICU1 fused to either cyan fluorescent protein (MICU1-CFP) or yellow fluorescent protein (MICU1-YFP) as previously described[49]. By performing a concentration response curve for Ca$^{2+}$-induced rearrangement of MICU1 multimers in HeLa cells, an EC$_{50}$ of 4.0 (3.3–5.0) μM was obtained (Fig. 5a) The siRNA-mediated reduction of PRMT1 did not change the sensitivity of MICU1 to Ca$^{2+}$ (EC$_{50}$ = 3.5 (2.6–4.6) μM; Fig. 5a). However, knockdown of UCP2/3 in this particular cell type strongly reduced Ca$^{2+}$ sensitivity of MICU1 and yielded a 4.6-fold increase in the EC$_{50}$ of Ca$^{2+}$-triggered MICU1 rearrangement to 18.5 (14.3–23.8) μM (Fig. 5b). Importantly, in PRMT1-depleted HeLa cells, knockdown of UCP2 had no effect on the Ca$^{2+}$ sensitivity of MICU1 rearrangement (EC$_{50}$ = 3.8 (3.0–4.8) μM; Fig. 5c). In contrast to the wild-type MICU1, the Ca$^{2+}$ sensitivity of the non-methylated MICU1-K expressed in HeLa cells was independent from the presence of UCP2/3 (with UCP2/3: EC$_{50}$ = 2.1 (1.6–2.9) μM Ca$^{2+}$; UCP2/3 knockdown EC$_{50}$ = 3.0 (2.3–3.9) μM Ca$^{2+}$ (Fig. 5d). These data are in line with the ITC data and confirm that methylation of MICU1 at position 455 yields a reduced Ca$^{2+}$ sensitivity of MICU1 that hampers MICU1 rearrangement and, thus, activation of MCU. The differences between the Ca$^{2+}$ affinities obtained with purified proteins in ITC experiments and that found in permeabilized cells (MICU1 FRET as well as mitochondrial Ca$^{2+}$ signalling) might be due to the fact that purified proteins were measured in solution and not embedded in membranes. Nevertheless, these data emphasize that the Ca$^{2+}$-dependent activation of mitochondrial Ca$^{2+}$ uniport, which is under the control of methylated MICU1, requires much higher Ca$^{2+}$ levels and exhibits less activity on identical Ca$^{2+}$ elevations (Fig. 3a). However, in the presence of UCP2/3, the Ca$^{2+}$ sensitivity of methylated MICU1 and, hence, the normal activity of the mitochondrial Ca$^{2+}$ uptake machinery is re-established.

**UCP2 ensures mitochondrial Ca$^{2+}$ uptake in cancer cells**. We observed UCP2/3 dependence of mitochondrial Ca$^{2+}$ in two cancer-type cell lines HeLa and Ea.hy926 (Fig. 1a,b), whereas independency of mitochondrial Ca$^{2+}$ uptake from UCP2 was observed in the respective cells from non-cancer tissue (Fig. 1c). Hence, the human breast cancer cell line MCF-7 was also found to exhibit UCP2-dependent mitochondrial Ca$^{2+}$ uptake (Fig. 5e). These findings may suggest that an engagement of UCP2/3 ensures mitochondrial Ca$^{2+}$ uptake activity under conditions of increased PRMT1 activity in certain cancers.

**Discussion**
Our findings demonstrate that the mitochondrial Ca$^{2+}$ uptake machinery is under the control of posttranslational protein modification by PRMT1. This protein arginine methyl transferase mediates methylation of MICU1 at position 455. Methylation of MICU1 results in a reduced Ca$^{2+}$ sensitivity for protein rearrangement and, thus, a reduced mitochondrial Ca$^{2+}$ uptake. Moreover, our data highlight that UCP2/3 function as a unique regulator of methylated MICU1 that becomes fundamental for mitochondrial Ca$^{2+}$ uniport under conditions of elevated PRMT1 activity (Fig. 6).

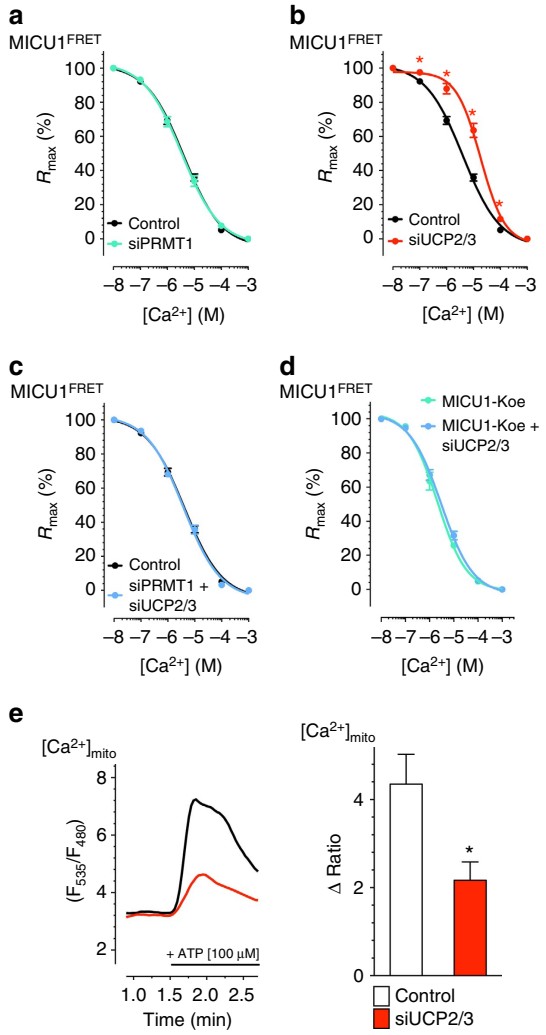

**Figure 5 | Methylation reduces sensitivity of MICU1 for rearrangement induced by Ca$^{2+}$.** (**a**) Concentration response curves of the Ca$^{2+}$-induced reduction of the MICU1-FRET ratio in HeLa cells treated with either control siRNA (black curve; $n = 20/20$) or in combination with siRNA against PRMT1 (light green curve; $n = 17/17$). Cells were treated with 3 μM ionomycin for 5 min and titrated with different concentrations of Ca$^{2+}$. Data expressed as mean ± s.e.m. (**b**) Concentration response curves of the Ca$^{2+}$-induced reduction of the MICU1-FRET ratio signal in HeLa cells with (red curve; $n = 17/17$) depletion of UCP2/3 or (**c**) combined depletion of PRMT1 and UCP2/3 (light blue curve; $n = 7/7$) in comparison with control siRNA (black curve; $n = 20/20$). Cells were treated with 3 μM ionomycin and different concentrations of Ca$^{2+}$. (**d**) Concentration response curves of Ca$^{2+}$-induced reduction of MICU1-FRET ratio signal in HeLa cells overexpressing MICU1-K mutant with (light blue curve; $n = 6/6$) or without (light green curve; $n = 6/6$) depletion of UCP2/3. (**e**) Left panel: representative curves showing mtCa$^{2+}$ ratio signals over time in response to 100 μM ATP in Ca$^{2+}$-free solution in MCF-7 cells with (red curve) or without (black curve) siRNA against UCP2/3. Right panel: bars show maximal Δ ratio signals of IP$_3$ generating agonist stimulation of control cells (white column: $n = 23/7$) or cells depleted of UCP2/3 (red column: $n = 26/8$). Numbers indicate the numbers of cells/independent repeats. *$P < 0.05$ versus control using the unpaired Student's $t$-test.

Although several core proteins of the mitochondrial uniporter complex could be identified so far (for reviews, see refs 12–14), it remains elusive how UCP2/3 interact functionally with mitochondrial Ca$^{2+}$ uptake. Our findings that an inhibition of PRMT1-mediated methylation by either the methylation inhibitor

Adox[42], the selective PRMT1 inhibitor AMI-1 (ref. 43) or knockdown of PRMT1 but not that of PRMT2, PRMT3, PRMT4 and PRMT6 (we excluded PRMT5, PRMT7 (ref. 50) and PRMT9 (ref. 51), as they are symmetrical dimethylarginine transferases, and PRMT8 because of its unique expression in the brain[52]) vanished the inhibitory effect of UCP2/3 knockdown on mitochondrial Ca$^{2+}$ uptake in HeLa cells pointing to a posttranslational protein modification as prerequisite for an engagement of UCP2/3 in mitochondrial Ca$^{2+}$ uptake. This assumption was further supported by our data showing that an introduction of PRMT1 in short-term cultured HUVEC cells turn mitochondrial Ca$^{2+}$ uptake from UCP2/3-independent to an UCP2/3-dependent process. Moreover, these data illustrate the transition of the mitochondrial Ca$^{2+}$ uptake from an UCP2/3-sensitive towards an UCP2/3-insensitive phenomenon in one individual cell and *vice versa*. Hence, as we did not find UCP2/3 dependency of mitochondrial Ca$^{2+}$ uptake in other cell lines (for example, human neuroblastoma cells) but could introduce UCP2/3 dependence of MCU activity in short-term cultured HUVEC cells by PRMT1, we speculate that the contribution of UCP2/3 in mitochondrial Ca$^{2+}$ uptake rather depends on PRMT1 activity than whether or not the cell is immortalized. Nevertheless, as cancer is often associated with enhanced PRMT activity[53], we cannot exclude that cell immortalization is associated with an UCP2/3 dependency of mitochondrial Ca$^{2+}$ uptake.

As a consequence of PRMT1-mediated protein methylation, our data show that the mitochondrial Ca$^{2+}$ uptake got less sensitive to Ca$^{2+}$. Although the reported shift from 5 to 14 μM Ca$^{2+}$ appears small, it matches the reported Ca$^{2+}$ concentrations of ∼4–16 μM that are established on intracellular Ca$^{2+}$ release in the ER–mitochondria junction[48]. Thus, we hypothesize that a PRMT1-mediated posttranslational modification of one or more of the core proteins of the mitochondrial Ca$^{2+}$ uniporter complex strongly affects the activity of mitochondrial Ca$^{2+}$ uptake. Hence, as in the presence of UCP2, the sensitivity of the mitochondrial Ca$^{2+}$ uptake appeared unaffected by PRMT1, our data indicate that UCP2 either prevents the activity of PRMT1 or counteracts the effect of PRMT1 and, thus, normalizes the Ca$^{2+}$ sensitivity of the mitochondrial Ca$^{2+}$ uptake.

Interestingly, we found an exclusive methylation of MICU1 but not of MCU and UCP2. Notably, methylation of MICU1 by PRMT1 was independent from the expression of UCP2/3. Thus, a direct effect of UCP2/3 on PRMT1 activity or MICU1 methylation appears unlikely. Therefore, we concentrated on MICU1 as the most important gatekeeper of the MCU complex[15,16]. The gatekeeping function of MICU1 on the mitochondrial Ca$^{2+}$ uniporter complex is controlled by binding of Ca$^{2+}$ to the EF-hand domains at the C-terminal end of MICU1 (refs 1,16,44). This part of the MICU1 protein extends into the intermembrane space[47,54]. This typography of MICU1 and its *in silico*-predicted methylation sites let us speculate that the arginine at position 455 might represent an important methylation site. The prediction was confirmed by methylation assays using the aDMA antibody in which wild-type MICU1 but neither the MICU1-K mutant nor the C terminus lacking MICUΔ$^{1-444}$ mutant were methylated by PRMT1. Notably, it is still possible that the arginine at position 455 of MICU1 is not the site of methylation but is required as an interaction/binding site for PRMT1 to get sufficient binding for catalysis. However, our findings that the MICU1-K mutant that mimics non-methylated MICU1 established an UCP2/3-independent mitochondrial Ca$^{2+}$ uptake despite high PRMT1 activity, whereas the MICU1-F mutant that mimics methylated MICU1 installed an UCP2/3-dependent MCU activity independently of the activity of PRMT1, favour our hypothesis that PRMT1-mediated methylation of arginine at position 455 of MICU1 is essential for the engagement of UCP2/3 in

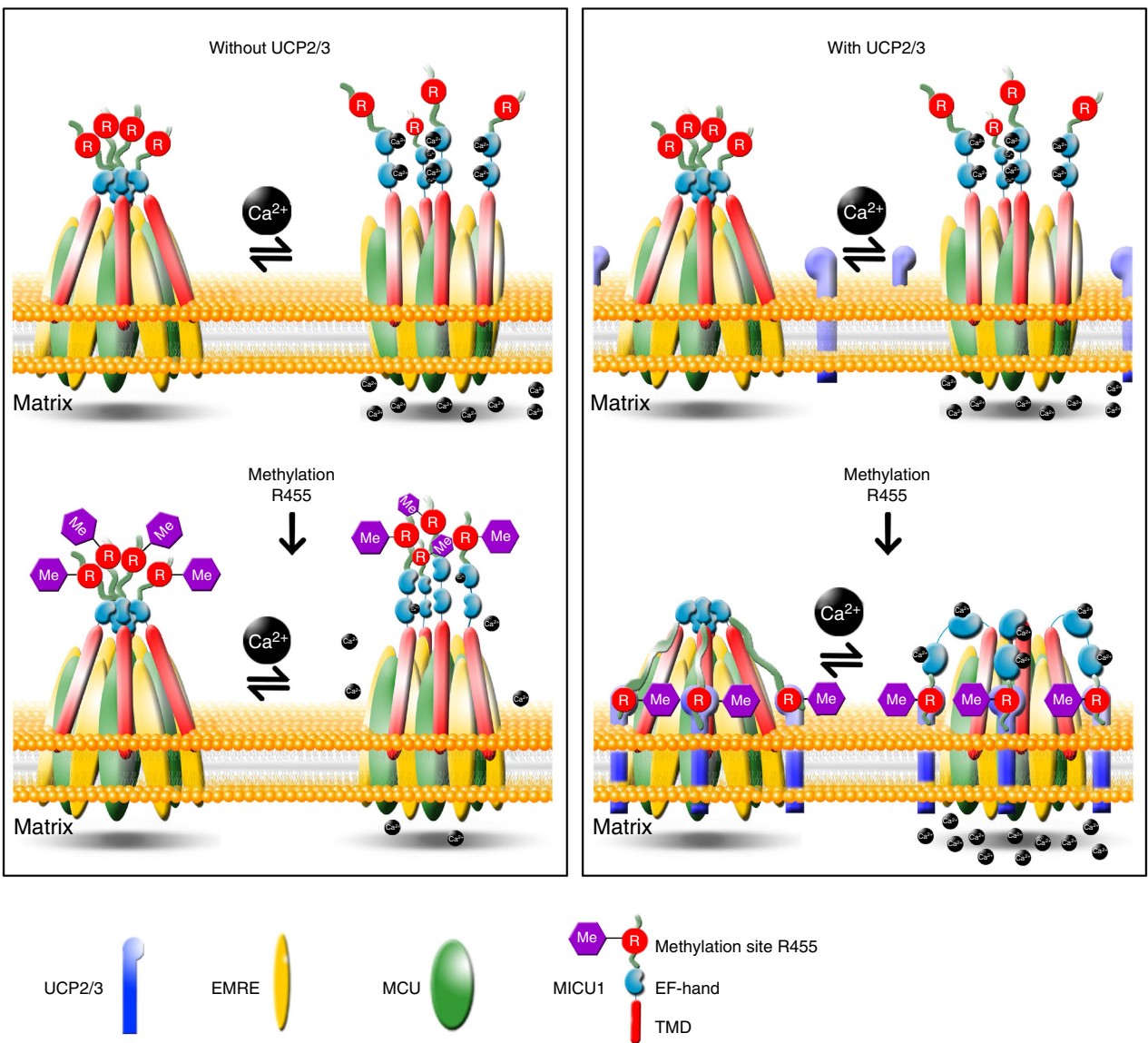

**Figure 6 | Schema of the engagement of UCP2/3 to mitochondrial Ca$^{2+}$ uptake under conditions of elevated PRMT1 activity.** Left panel: in the absence of PRMT1 activity, mitochondrial Ca$^{2+}$ uptake is under the control of MICU1 (upper part). In the lack of UCP2, methylation of MICU1 by PRMT1 yields desensitization to Ca$^{2+}$, thus, resulting in reduced mitochondrial Ca$^{2+}$ uptake under conditions of high PRMT1 activity (lower part). Right panel: although UCP2/3 do not contribute to mitochondrial Ca$^{2+}$ uptake under conditions with no PRMT1 activity (upper part), upon increased methylation activity, UCP2/3 ensure Ca$^{2+}$ sensitivity of MICU1 and, thus, the mitochondrial Ca$^{2+}$ uptake complex stabilization (lower part), while mitochondrial Ca$^{2+}$ uptake remains unchanged despite high PRMT1 activity.

mitochondrial Ca$^{2+}$ uptake. Moreover, although we cannot exclude the possibility that other not tested protein(s) (for example, EMRE, MCUb, MICU2, MCUR1, or so far unknown components) of the MCU complex also serves as target of PRMT1, our data highlight methylation of MICU1 as being the essential step for the engagement of UCP2/3 in the regulation of mitochondrial Ca$^{2+}$ uptake.

Very recently we demonstrated that a cytosolic/intermembrane Ca$^{2+}$ elevation, which results in binding of Ca$^{2+}$ to the EF-hand domains of MICU1, triggers a clear rearrangement of MICU1 multimers independently from the mitochondrial matrix Ca$^{2+}$ concentration, mitochondrial membrane potential, and the level of expression of either MCU or EMRE[49]. This Ca$^{2+}$-induced rearrangement of MICU1 proteins is, however, essential to allow an efficient mitochondrial Ca$^{2+}$ uptake via the MCU/EMRE channel. In line with the reduction of sensitivity of the mitochondrial Ca$^{2+}$ uptake by PRMT1-mediated methylation

of MICU1, *in vitro* ITC revealed a strongly reduced sensitivity for Ca$^{2+}$ to trigger de-oligomerization of MICU1 on methylation of MICU1 at position 455. Excitingly, the range of Ca$^{2+}$ desensitization of MICU1 oligomerization by methylation in the ITC measurements perfectly matches that on mitochondrial Ca$^{2+}$ uptake and of the Ca$^{2+}$-induced MICU1 rearrangement. Hence, as mutation at position 455 of the MICU1 prevented Ca$^{2+}$ desensitization in the ITC measurements as well as in our MICU1 FRET experiments, we conclude that PRMT1-mediated methylation of the arginine at position 455 of MICU1 is responsible for the loss of Ca$^{2+}$ sensitivity of MICU1.

Notably, our ITC experiments further demonstrate that UCP2 normalizes the reduced Ca$^{2+}$-induced oligomerization of methylated MICU1, while UCP2 did not bind to non-methylated MICU1. These findings are in line with our data obtained in the MICU1–MICU1 FRET experiments where UCP2 re-established normal Ca$^{2+}$ sensitivity of the Ca$^{2+}$-induced rearrangement

of methylated MICU1, whereas UCP2 had no effect on the rearrangement of non-methylated MICU1.

All together, our data presented herein support our previous findings on an engagement of UCP2/3 in the regulation of mitochondrial $Ca^{2+}$ uptake and prove UCP2/3 as essential sensitizer of methylated MICU1 and, thus, being fundamental for establishing MCU activity under conditions where PRMT1 is upregulated (Fig. 5f).

Mitochondrial $Ca^{2+}$ uptake is a key regulator for the organelle's functions[55] and dysfunctions[56]. Accordingly, our findings on a posttranslational regulation of MICU1 by PRMT1 that results in a decreased $Ca^{2+}$ sensitivity of methylated MICU1 and, subsequently, a reduced activity of mitochondrial $Ca^{2+}$ uptake, opens a new aspect of tuning mitochondrial activity in the orchestration of cellular (energy) metabolism. So far, little is known on the modulation of the activity of the MCU complex. On the transcriptional level, the expression patterns of the proteins involved in the mitochondrial $Ca^{2+}$ uptake were described to shape the activity of neuronal cell types[57,58] and cancer growth[59]. Hence, MCU expression was found to be under the $Ca^{2+}$-CREB pathway in chicken DT40 B lymphocytes[60]. Considering the importance of MICU1 for regulating MCU activity[16,61], this protein appears to be a very crucial target for posttranslational modification(s). Accordingly, the essential oxidoreductase Mia40/CHCHD4 has been described to bind to MICU1 and to facilitate its dimerization with MICU2 (ref. 62) and this study points to PRMT1 and UCP2/3 as important regulator of MCU activity. The reason(s) of the alternate PRMT1 activities in different cell types/cultures that would explain the contrary results regarding the involvement of UCP2/3 in mitochondrial $Ca^{2+}$ uptake are so far unknown but might be due to differences in medium compositions, growth rates, age, or passage number of the cell (line) used and/or cell handling before experiments. However, tissue-specific variations in PRMT1 activity[63] might contribute to differences in the activity of mitochondrial $Ca^{2+}$ uniporter between various tissues[64]. Moreover, an enhanced PRMT1 activity as a key mechanism for a metabolic adaptation has been associated with ageing[40] and several severe diseases such as amyotrophic lateral sclerosis[36] and cancer[50,65]. However, an increased PRMT1 activity also results in methylation of MICU1 and a reduced mitochondrial $Ca^{2+}$ uptake/activity would counteract the elevated demand of cancer cells on $Ca^{2+}$-triggered ATP production in the mitochondria. By using UCP2/3, cancer cells may be able to counteract the impact of an essential PRMT1 activity to mitochondrial $Ca^{2+}$ uptake and, thus, re-establish activity of mitochondrial $Ca^{2+}$ uptake to achieve $Ca^{2+}$-triggered activation of ATP production. Notably, increased UCP2 expression has been reported in several cancer types[53]. In line with this hypothesis, UCP2 has been identified as survival factor in breast cancer cells[66], a cancer type that has been shown to be associated with elevated PRMT1 activity[50,67]. Consistently, our data in human breast cancer cell line (MCF-7) approve the existence of an UCP2/3 dependence of mitochondrial $Ca^{2+}$ uptake fueling the hypothesis that UCP2 might play an important role in the energy balance of certain cancer cells. Obviously, further studies are required to explore the interrelation between UCP2/3 expression and PRMT1 activity in cancer and to reveal whether or not this function of UCP2/3 is causally linked to altered UCP2/3 expression levels in various cancer cells.

The mystery of the occasionally observed engagement of UCP2/3 in the regulation of MCU activity appears to be solved. Our findings demonstrate that the mitochondrial $Ca^{2+}$ uptake machinery is under the control of posttranslational protein modification by PRMT1 that methylates MICU1 at position 455. Methylation of MICU1 yields reduced $Ca^{2}$ sensitivity for protein rearrangement and, thus, reduced mitochondrial $Ca^{2+}$ uptake.

UCP2 is described as unique regulator of methylated MICU1 that normalizes the $Ca^{2+}$ sensitivity of methylated MICU1 and, consequently, allows the re-establishment of mitochondrial $Ca^{2+}$ uptake. Accordingly, although originally not involved in the regulation of mitochondrial $Ca^{2+}$ uptake, UCP2/3 get engaged as sensitizer of methylated MICU1 preserving the activity of mitochondrial $Ca^{2+}$ uptake despite the reduced $Ca^{2+}$ sensitivity of methylated MICU1 (Fig. 6).

## Methods

**Chemicals and buffer solutions.** Cell culture materials were obtained from PAA Laboratories (Pasching, Austria). Histamine, BHQ, and EGTA were purchased from Sigma-Aldrich (Vienna, Austria). Before experiments, cells were washed and maintained for 20 min in a HEPES-buffered solution containing 138 mM NaCl, 5 mM KCl, 2 mM $CaCl_2$, 1 mM $MgCl_2$, 1 mM HEPES, 2.6 mM $NaHCO_3$, 0.44 mM $KH_2PO_4$, 0.34 mM $Na_2HPO_4$, 1 mM D-glucose, 0.1% vitamins, 0.2% essential amino acids and 1% penicillin–streptomycin, pH adjusted to 7.4 with NaOH or HCl. During the experiments, cells were perfused with a $Ca^{2+}$-containing buffer, which consisted of 145 mM NaCl, 5 mM KCl, 2 mM $CaCl_2$, 1 mM $MgCl_2$, 10 mM D-glucose, and 10 mM HEPES, pH adjusted to 7.4, or with a $Ca^{2+}$-free buffer, in which $CaCl_2$ was replaced by 1 mM EGTA.

**Cells and cell handling.** HeLa S3 (ATCC CCL-2.2) and Ea.hy926 cells (provided by Dr C.J.S. Edgell, University of North Carolina, Chapel Hill, NC, USA) were grown in DMEM medium (Sigma-Aldrich) containing 10% fetal bovine serum (PAA Laboratories), 100 U ml$^{-1}$ penicillin and 100 µg ml$^{-1}$ streptomycin, and 2 mM glutamine (Gibco, Life Technologies, Vienna, Austria). By routine, long-term cultured cells were tested for mycoplasma contamination and were tested to be free of mycoplasma. For $Ca^{2+}$ imaging, cells were plated on 30 mm glass coverslips and transiently transfected at 60–80% confluence with 1.5 µg plasmid DNA encoding the appropriate sensor alone or with 100 µM siRNA using 2.5 µl of TransFast transfection reagent (Promega, Madison, WI, USA) in 1 ml of serum- and anti-biotic-free medium. Cells were maintained in a humidified incubator (37 °C, 5% $CO_2$, 95% air) for 16–20 h. Afterwards, transfection mix was replaced by full culture medium. All experiments were performed 48 h after transfection. siRNAs were obtained from Microsynth (Balgach, Switzerland) and their sequences (5′–3′) were as follows: human UCP2 siRNA: 5′-GCA CCG UCA AUG CCU ACA A dTdT-3′; human UCP3 siRNA: 5′-GGA ACU UUG CCC AAC AUC A dTdT-3′; human PRMT1 siRNA: 5′-CGU CAA AGC CAA CAA GUU A dTdT-3′; human MCU siRNA-1: 5′-GCC AGA GAC AGA CAA UAC U dTdT-3′; human MCU siRNA-2: 5′-GGA AAG GGA GCU UAU UGA A dTdT-3′; human UCP2 siRNA: 5′-UGU CGC UCG UAA UGC CAU U dTdT-3′; human PRMT2 siRNA: 5′-CCC UGA CGG AUA AAG UCA U dTdT-3′; human PRMT3 siRNA: 5′-GCC UUG UAG CAG UGA GUG A dTdT-3′; human PRMT4 siRNA: 5′-GUA CAC GGU GAA CUU CUU A dTdT-3′; and human PRMT6 siRNA: 5′-CGG GAC CAG CUG UAC UAC G dTdT-3′. For detection primers, see Supplementary Table 3.

HUVEC cells were isolated from umbilical vein by scraping and afterwards cultured in endothelial cell growth medium (EGM-2) (Lonza, Basel, Switzerland). PAECs were harvested by scraping and cultured in DMEM containing supplements mentioned above. For imaging experiments, HUVEC cells and PAECs were seeded on poly-L-lysine-coated 30 mm glass coverslips in six-well plates, transiently transfected at 60–80% confluence with 100 µM siRNA using 2.5 µl of TransFast transfection reagent in 1 ml of serum- and antibiotic-free medium, and infected with BacMam 4mtD3cpv virus (Life Technologies, Vienna, Austria) following the CellLight protocol. Endothelial cells have been authenticated by factor VII and/or endothelial nitric oxide synthase (eNOS) staining. Experiments were performed 48 h after transfection and infection.

**mRNA isolation, real time and detection PCRs.** Total RNA was isolated using the PEQLAB total RNA isolation kit (Peqlab, Erlangen, Germany) and reverse transcription was performed in a thermal cycler (Peqlab) using a complementary DNA synthesis kit (Applied Biosystems, Foster City, CA). Expression of core constituents of mitochondrial $Ca^{2+}$ uptake machinery and PRMT1 in HeLa and Ea.hy926 cells, as well as in HUVECs and PAECs, was examined by RT–PCR. A QuantiFast SYBR Green RT–PCR kit (Qiagen, Hilden, Germany) was used to perform real time PCR on a LightCycler 480 (Roche Diagnostics, Vienna, Austria) and data were analysed by the REST Software (Qiagen). Relative expression of specific genes was normalized to human (QuantiTect; Qiagen) or porcine GAPDH (Supplementary Table 3) respectively, as a housekeeping gene. Primers (Supplementary Table 3) for real-time PCR were obtained from Invitrogen (Vienna, Austria).

For detecting the presence of the different PRMT isoforms in HeLa S3, Ea.hy926, HUVEC and in PAEC cells, specific primers (Invitrogen) were designed (Supplementary Table 3) and used for PCR. mRNA was isolated from the cell lines and transcribed to cDNA, which was together with the GoTaq Green Master Mix (Promega; Mannheim, Germany) used to set up the PCR reaction. The thermocycler (VWR, Vienna, Austria) was programmed as follows: initial denaturation at 95 °C for 10 min, followed by 35 cycles of denaturation for 45 s at 95 °C, annealing for

45 s at 58 °C and elongation for 45 s at 72 °C, and a final elongation step of 10 min at 72 °C. After the PCR samples were applied on a 1.5% agarose gel (PeqLab) and electrophoresis was run at 70 V for 45 min.

**FRET measurements.** Dynamic changes in $[Ca^{2+}]_{mito}$ and $[Ca^{2+}]_{ER}$ were followed in cells expressing the 4mtD3cpv or D1ER[21]. Medium was removed and cells were kept in loading buffer containing 135 mM NaCl, 5 mM KCl, 2 mM CaCl₂, 1 mM MgCl₂, 10 mM Hepes, 2.6 mM NaHCO₃, 440 μM KH₂PO₄, 340 μM Na₂HPO₄, 10 mM D-glucose, 0.1% vitamins, 0.2% essential amino acids, and 1% penicillin–streptomycin, pH adjusted to 7.4. Single-cell measurements were performed on a Zeiss AxioVert inverted microscope (Zeiss, Göttingen, Germany) equipped with a polychromator illumination system (VisiChrome, Visitron Systems, Puchheim, Germany) and a thermoelectric-cooled CCD (charge-coupled device) camera (Photometrics CoolSNAP HQ, Visitron Systems). Transfected cells were imaged with a × 40 oil-immersion objective (Zeiss). Excitation of the fluorophores was at 440 ± 10 nm (440AF21, Omega Optical, Brattleboro, VT) and emission was recorded at 480 and 535 nm using emission filters (480AF30 and 535AF26, Omega Optical) mounted on a Ludl filterwheel. Devices were controlled and data were acquired by VisiView 2.0.3 (Visitron Systems) software and analysed with GraphPad Prism version 5.00 for Windows (GraphPad Software, San Diego, CA). Results of FRET measurements are shown as $(R_i − \text{Background}) + [(R_i − \text{Background}) − (R_0 − \text{Background})]$ (where $R_0$ is the basal ratio) to correct for photobleaching and/or photochromism.

**MICU1 FRET measurements.** MICU1 FRET measurements were performed as described previously[49]. In brief, MICU1-eCFP and MICU1-citrine co-expressing HeLa cells were illuminated at 440 nm and the FRET ratio was calculated from the emitted light at 535 and 480 nm, respectively. Dissociation constants ($K_d$) of MICU1 FRET rearrangement were determined in diverse knockdown and overexpression experiments by the application of various $Ca^{2+}$ concentrations ranging from 0.1 to 1,000 μM, to permeablized HeLa cells.

**Cytosolic $Ca^{2+}$ imaging.** Cells were incubated with 2 μM Fura-2/AM (TEFLabs, Austin, TX) in loading buffer for 30 min and were alternately illuminated at 340 and 380 nm, whereas fluorescence emission was recorded at 510 nm[21]. Results of Fura-2/AM measurements are shown as the ratio of $F_{380}/F_{340}$.

**Cloning.** For engineering MICU1 mutants (MICU1-K and MICU1-F), an already existing MICU1 plasmid was amplified via PCR with primers containing the respective mutation sequence. For both mutants the forward primer 5′-ACG-GATCCATGTTTCGTCTGAACTCAC-3′ with a BamHI restriction site was used. As the mutations are very close to the end of MICU1, we decided to take a long reverse primer with a EcoRI restriction site and the following sequence: 5′-AAGAATTCCTGTTTGGGTAAAGCGAAGTCCCAGGCAGTTTCCTGTGCACATTTCCACATGGCCTGCATGAGXXXAGTGAAACCCAT-3′, whereas XXX for MICU1-K is TTT and for MICU1-F AAA. The PCR fragment was cloned into a pcDNA3.1 ( + ) vector via BamHI and EcoRI sites already C-terminally containing the sequence coding for the red fluorescent protein mCherry. MICU1-CFP and MICU1-YFP were constructed by amplifying the coding sequences of human MICU1 (hMICU1, NM_006077.3) without the stop codon from a HeLa cDNA by PCR. The following primers (Invitrogen) were used: forward: 5′-ACGGAT CCACCATGTTTCGTCTGAACTCAC-3′ and reverse: 5′-ACGAATTCCTGTTT GGGTAAAGCGAAGTCC-3′. Subsequently, the PCR fragment was cloned into pcDNA3.1 ( + ) vector using BamHI and EcoRI sites. The coding sequences of enhanced CFP and citrine (YFP) were amplified with the same primers: forward: 5′-AAGAATTCATGGTGAGCAAGGGCGAGGAG-3′ and reverse: 5′-CCTCT AGAACTTGTACAGCTCGTCCATGC-3′. Each PCR product was C-terminally fused to hMICU1 using the EcoRI and XbaI sites[49].

**Immunoprecipitation and western blot analysis.** Cells were lysed with RIPA buffer (Biorad formulation), scraped, and sonicated (80% amplitude, 15 s). Protein A/G plus agarose beads (sc2003, 25 μl, Santa Cruz Biotechnology, Heidelberg, Germany) were washed twice with 1 ml IP washing buffer (Abcam formulation) and incubated with 1 μg anti-flag antibody (F1804, Sigma-Aldrich) for 2 h at 4 °C. Protein lysates were added and incubated with the coated beads overnight at 4 °C on a shaker. After three washing cycles with 1 ml IP-washing buffer NuPAGE LDS sample buffer (Fisher Scientific, Vienna, Austria) and 1 μl of NuPAGE Sample Reducing Agent (Fisher Scientific) was added to the beads-lysate mix, heated for 10 min at 70 °C and 800 r.p.m. and, afterwards, directly loaded onto an SDS–PAGE gel.
Western blots were performed following standard protocols. Briefly, samples were heat denatured in 1 × Laemmli sample buffer and resolved in an 10% SDS–PAGE gel and Mini-PROTEAN TGX precast gels (#4561094, BioRad) in parallel to PageRulerTM Plus Prestained Protein Ladder (Fisher Scientific) and then transferred onto a polyvinylidene difluoride membrane (Millipore, Vienna, Austria). To exclude unspecific bands due to impurities of immunoprecipitation, normal mouse IgG (sc-2025, Santa Cruz Biotechnology) and a control lysate (also undergone immunoprecipitation procedure) were subjected to western

blotting. Proteins were detected using the following antibodies: MICU1 (sc-16021, 1:200, Santa Cruz Biotechnology, and #12524, 1:1,000, from New England Biolabs, Inc., Frankfurt, Germany), MCU (#14997, 1:1,000, Santa Cruz Biotechnology), UCP2 (ab67241. 1:1,000, Abcam, Cambridge, UK), Flag (F1804, 1:5,000, Sigma-Aldrich) and β-actin (sc-47778, 1:500, Santa Cruz Biotechnology). The methylation status was detected by the asymmetric dimethyl arginine motif aDMA antibody (#13522, 1:1,000, New England Biolabs, Inc.) and the anti-mono and dimethyl arginine antibody (MMA/SDMA) (ab412, 1:1,000, Abcam). The secondary horseradish peroxidase antibodies, anti-mouse, anti-rabbit (PI-2,000 and PI-1,000, both 1:1,000, Vector Laboratories, Burlingame, USA), anti-goat (sc-2020, 1:4,000, Santa Cruz Biotechnology), and VeriBlot for IP antibody (ab131366, 1:250, Abcam), were used and visualized with super signal west pico luminol/enhancer developing solution (Fisher Scientific). As a control for the IgG heavy and light chains, mouse normal IgG (sc-2025, 2 μl, Santa Cruz Biotechnology) was used.

**Co-localization analysis of ER and mitochondria.** Hela cells were transfected with D1ER and PRMT1 construct or siRNA against PRMT1. After 2 days, cells were stained for 10 min with 200 nM MitoTracker Red CMXRos and imaged directly with 488 nm laser excitation (D1ER) and a 561 nm laser (MitoTracker Red CMXRos) with a CFI SR Apochromat TIRF × 100 oil (NA1.49) objective mounted on a Nikon-Structured Illumination Microscopy (SIM) System equipped with an Andor iXon3 EMCCD camera. Three-dimensional (3D) SIM was used in both channels with 30 ms exposure time per image. SIM images where reconstructed using Nis-Elements (Nikon). Images were background corrected with an ImageJ-Plugin (Mosaic Suite, background substractor). Further, the ImageJ/Fiji tool coloc2 was used to determine Pearson and Mander's coefficients, whereby images were thresholded with Costes automatic threshold to determine Mander's coefficient (overlap of mitochondria with ER).

**Co-localization of MICU1 constructs to mitochondria.** Hela cells were transfected with tagged MICU1 constructs (MICU1-WT/MICU1-K/MICU1-F). After 2 days, cells were stained for 10 min with 200 nM MitoTracker Red CMXRos and imaged directly. High-resolution images of cells were recorded by using a confocal spinning disk microscope (Axio Observer.Z1 from Zeiss) equipped with × 100 objective lens (Plan-Fluor × 100/1.45 Oil, Zeiss), a motorized filter wheel (CSUX1FW, Yokogawa Electric Corporation, Tokyo, Japan) on the emission side, AOTF-based laser merge module for laser line (405, 445, 473, 488, 561, and 561 nm (Visitron Systems), and a Nipkow-based confocal scanning unit (CSU-X1, Yokogawa Electric Corporation). The MICU1-YFP constructs and Mitotracker Red CMXRos were excited with 488 and 561 laser lines, respectively, and emission was acquired with a charged CCD camera (CoolSNAP-HQ, Photometrics, Tucson, AZ, USA). The software VisiView acquisition software (Universal Imaging, Visitron Systems) was used to acquire the imaging data. Images were background corrected with an ImageJ-Plugin (Mosaic Suite, background substractor).

**The co-localization of MICU1 constructs with MCU and EMRE.** Hela cells were transfected with MICU1-YFP constructs (MICU1-WT/MICU1-F/MICU1-K) and MCU-mCherry or EMRE-mCherry. After 2 days, cells were imaged directly using a confocal spinning disk microscope. The MICU1-YFP constructs (MICU1-WT/MICU1-F/MICU1-K) and MCU-mCherry or EMRE-mCherry were excited with 488 and 561 laser lines, respectively. Images were background corrected with an ImageJ-Plugin (Mosaic Suite, background substractor). Co-localization was measured as Pearson, Manders1 and Manders2 co-localization coefficients using the Coloc2-tool in ImageJ. For Manders 1/2 coefficient, the automated Costes threshold regression was chosen. Manders 1 defines the fraction of YFP-tagged MICU1-WT, MICU1-K or MICU1-F with EMRE- or MCU-mCherry and Manders 2, the fraction of EMRE- or MCU-mCherry overlapping with MICU1-WT, MICU1-F, and MICU1-K.

**Effects of MICU1 constructs on mitochondrial morphology.** Hela cells were transfected with MICU1-YFP constructs (MICU1-WT/MICU1-F/MICU1-K). After 2 days, cells were stained for 10 min with 200 nM MitoTracker Red CMXRos and transfected cell were imaged directly with 488 nm laser (YFP) and a 561 nm laser excitation (MitoTracker Red CMXRos). Whole-cell Z-stacks of the same cells were performed with 561 nm excitation with 200 nm Z-step size. 3D stacks of Micu1-YFP (MICU1-WT/MICU1-F/MICU1-K)-transfected cells were analysed with ImageJ. Z-stacks were background corrected with an ImageJ-Plugin (Mosaic Suite, background substractor) and auto thresholded by using whole stack histogram and segmented using the 3D-Manager of ImageJ. By using the tool '3D-ellipsoid fitting' for segmented mitochondria, elongation and flatness factors were determined. The plugin '3D Geometrical Measure' was used to measure the volume and surface area of segmented mitochondria. The 3D formfactor (FF) was determined with following equation:

$$FF = \sqrt[3]{\frac{36 \cdot \pi \cdot V^2}{S^3}}$$

where $S$ stands for the surface area and $V$ for the volume of the segmented mitochondria. Lower values show higher mitochondrial branching and higher values less complexity.

**Protein expression and purification.** For the *in vitro* interaction studies, the *Escherichia coli* codon optimized genes of human MICU1 97-444 and 97-476 were cloned into pETM11 expression vectors (including a His$_6$, protein A tag and a tobacco etch virus (TEV) protease cleavage site). The MICU1 R455K mutant was generated using the QuikChange Site-Directed Mutagenesis Kit. Expression plasmids were transformed into *E. coli* BL21(DE3) and grown in standard lysogeny broth medium. Protein synthesis was induced by addition of 1 mM isopropyl β-D-1-thiogalactopyranoside at OD$_{600}$ ∼ 0.8. After protein expression for 18 h at 16 °C, cells were collected by centrifugation, re-suspended in lysis buffer consisting of 20 mM Tris (pH 7.0), 300 mM NaCl, 10 mM imidazol, 5 mM β-mercaptoethanol, 10 mM MgCl$_2$ and 0.02% (w/v) NaN$_3$, and lysed by sonication in the presence of lysozyme, Benzonase Nuclease (Sigma-Aldrich) and Protease Inhibitor Mix HP (Serva Electrophoresis). The cell lysate was cleared by centrifugation at 20,000 $g$ for 45 min at 4 °C. The supernatant was loaded onto a HisTrap FF column (GE Healthcare Life Sciences) attached to an Akta Pure FPLC system (GE Healthcare) and washed with ten column volumes of wash buffer consisting of 20 mM Tris pH 7.0, 300 mM NaCl, 20 mM imidazol, 5 mM β-mercaptoethanol, 10 mM MgCl$_2$ and 0.02% (w/v) NaN$_3$. The sample was eluted with the same buffer containing 400 mM imidazol. Arginine (50 mM) were added after elution to improve long-term stability of the protein.

Rat PRMT1 with a deletion of the first ten amino acids was produced in *E. Coli* Bl21 (DE3) cells harbouring a pET28b-PRMT1 vector containing a amino-terminal His$_6$-tag (provided by Dr Dorothee Dormann, LMU, Munich). This truncated version is identical to human PRMT1 (uniprot ID: Q99873-3), except a single amino acid exchange (H161 is Y in human), and represents the vast majority of expressed sequence tags[68]. Cells were freshly inoculated and grown overnight in 20 ml of lysogeny broth medium at 37 °C. Next day, the culture was diluted to 1 litre and grown to an optical density of 0.8 at 37 °C. Protein expression was induced by addition of 0.5 mM isopropyl β-D-1-thiogalactopyranoside and incubation overnight at 19 °C. Next day, cell pellets were resuspended in lysis buffer (see above), sonicated and centrifuged for 1 h. The cell lysate was loaded on a pre-packed Ni Sepharose column (HisTrap FF, GE Healthcare) attached to an Akta Pure FPLC system (GE Healthcare) and equilibrated with a 50 mM Tris buffer pH 7.5, 150 mM NaCl, 2 mM tris(2-carboxyethyl)phosphine and 20 mM imidazole (purification buffer). Nonspecific proteins were removed by washing with ten column volumes of purification buffer. Protein elution was performed by the same buffer containing 500 mM imidazole.

**In vitro arginine methylation.** For PRMT1-dependent arginine methylation, the buffer was exchanged via PD-10 Desalting Columns (GE Healthcare Life Sciences) to methylation buffer consisting of 50 mM Na$_2$HPO$_4$/NaH$_2$PO$_4$ pH 8.0, 150 mM NaCl and 2 mM β-mercaptoethanol. For the methylation reaction, recombinant MICU1 (40 μM) and PRMT1 (14 μM) were mixed to final concentrations of 20 and 7 μM, respectively. S-adenosyl-L-methionine (New England Biolabs, Inc.; 33 mM) was added to a final concentration of 1 mM. Volumes of the methylation reaction were in the order of several millilitres and depending on the protein yield. The reaction was carried out overnight at room temperature. After methylation, the proteins were buffer exchanged into lysis buffer (20 mM Tris pH 7.0, 300 mM NaCl, 10 mM imidazol, 5 mM β-mercaptoethanol (BME), 10 mM MgCl$_2$ and 0.02% (w/v) NaN$_3$) using a HiPrep 26/10 desalting column attached to an Äkta pure FPLC system (GE Healthcare). Arginine (50 mM) was added to stabilize the protein. 2 (w/w)% TEV Protease were added and cleavage was performed overnight at 4 °C. After TEV cleavage, proteins were exchanged into lysis buffer without arginine (20 mM Tris pH 7.0, 300 mM NaCl, 10 mM imidazol, 5 mM BME, 10 mM MgCl$_2$ and 0.02% (w/v) NaN$_3$) using a HiPrep 26/10 Desalting column attached to an Äkta pure FPLC system (GE Healthcare) and then applied onto a Ni-NTA column (Ni-NTA agarose, Thermo Fisher) and the flow through containing cleaved MICU1 was collected. As the PRMT1 construct does not contain a TEV cleavage site, its His$_6$-tag is not cleaved and PRMT1 can be removed. The flow through was further concentrated to a final volume of 2 ml using 10 kD cutoff Amicon Ultra-15 ml centrifugal filter units (Millipore) in a centrifuge with Swinging-bucket rotor.

A size-exclusion step was performed to remove contaminants using a Superdex 75 16/600 column attached to an Äkta pure FPLC system and equilibrated in MICU1 ITC buffer (20 mM Tris pH 7.0, 500 mM NaCl, 50 mM Arg and 2 mM BME). After size-exclusion chromatography (SEC), MICU1 was concentrated to a 20 μM solution using a 10 kD cutoff Amicon Ultra-15 ml centrifugal filter unit (Millipore) with centrifugation steps of 5 min at 3,500 r.p.m. in a centrifuge with Swinging-bucket rotor.

**Isothermal titration calorimetry.** Binding affinities of MICU1 to CaCl$_2$ and to a synthetic IML2$^{UCP}$ peptide (residues 147–171 of human UCPs) were determined by ITC on a VP-ITC Microcal calorimeter (Microcal, Northhampton, USA) at 25 °C with 29 rounds of 10 μl injections (first injection 2 μl). All proteins/peptides were buffer exchanged to MICU1 buffer (20 mM Tris pH 7.0, 500 mM NaCl, 50 mM Arg and 2 mM BME) before ITC. The MICU1 concentration in the cell was 40 μM for all measurements. CaCl$_2$ and IML2$^{UCP2}$ concentration in the syringe was 500 μM. For measurement where CaCl$_2$ was titrated to MICU1/IML2$^{UCP2}$, we used a UCP concentration of 100 μM. The ITC data were analysed with the programme MicroCal Origin software version 7.0 and single site-binding model.

**Statistics.** Data shown represent the mean ± s.e.m. All experiments were performed at least three times with Ea.hy926 and HeLa, and for experiments with HUVECs and PAECs three different donors were used. For live-cell imaging, numbers indicate the numbers of cells/independent repeats. Statistical analyses were performed by using the unpaired Student's $t$-test and $P < 0.05$ was considered to be significant.

**Data availability.** The data that support the findings of this study are available from the corresponding author upon request.

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

## Acknowledgements

We thank Dr C.J.S. Edgell (University of North Carolina, Chapel Hill, NC, USA) for providing us with the Ea.hy926 cells. This work was funded by the Austrian Science Funds (FWF, DKplus W 1226-B18 and P 28529-B27). C.T.M. is a doctoral fellow within the doctoral programme Metabolic and Cardiovascular Disease (FWF, DKplus W 1226-B18) at the Medical University of Graz. W.P. is supported by the Austrian Academic Exchange Services (ÖAD) and is a doctoral fellow of the doctoral school Molecular Medicine at the Medical University of Graz. Microscopic equipment is part of the Nikon-Center of Excellence, Graz, which is supported by the Austrian infrastructure programme 2013/2014, Nikon Austria, Inc. and BioTechMed. T.M. was supported by the Bavarian Ministry of Sciences, Research and the Arts (Bavarian Molecular Biosystems Research Network), the German Research Foundation (Emmy Noether program MA 5703/1-1), the Center for Integrated Protein Science Munich (CIPSM) and the Austrian Science Fund (FWF: P28854).

## Author contributions

C.T.M.-S., W.P., A.I.B. and M.W.-W. performed experimental work and FRET measurements. M.W.-W., C.K. and E.E. cloned the constructs and performed western blots and immunoprecipitation. R.R. was responsible for cell culture and transfection. B.G. performed super-resolution microscopy and image data analyses. S.P. and S.S. cloned and purified recombinant proteins, and performed in vitro methylation assays. S.S. and T.M. carried out ITC experiments. T.M. designed MICU1 mutants and provided critical materials, and participated in the preparation of the manuscript. R.M. together with W.F.G. supervised research and project planning, performed data interpretation and prepared the manuscript. All authors discussed the results and implications, and commented on the manuscript at all stages.

## Additional information

**Competing financial interests:** The authors declare no competing financial interests.

