## [Peer review file · Nature Communications]

Reviewers' comments:

Reviewer #1 (Remarks to the Author):

I find this work to be extremely interesting and insightful, albeit not necessarily for the message the authors intended to convey. I do believe that the title "PRMT1-mediated methylation of MICU1 determines the UCP2/3-dependency of mitochondrial Ca²⁺ uptake" properly describes their findings for immortalized cells. The crucial finding in the view of the reviewer is that the same does not hold for freshly harvested, non-immortalized HUVEC cells. Reliance on mitochondrial functions of immortalized cells is dramatically different from that of primary cultures and cells in situ. In fact, this paper may offer a resolution for many of the controversies regarding this issue (role of UCPs in calcium transport).

I would support the paper, if the above reasoning would be accepted by the authors. They should adjust the title to reflect the findings in immortalized cells. Of course, to strengthen the point I am raising here, they could do some more work on non-immortalized cells to confirm or disconfirm their current possible conclusions.

Reviewer #2 (Remarks to the Author):

The manuscript entitled "PRMT1-mediated methylation of MICU1 determines the UCP2/3-dependency of mitochondrial Ca²⁺ uptake" by Sokolowski et al., is a follow up of the previous works by the same group. The authors have focused on showing evidence for the UCP2/3 dependency of mitochondrial Ca²⁺ uptake via the methylation of MICU1 by PRMT1. From the biochemical point of view the authors argue the dependency of mitochondrial Ca²⁺ uptake by UCP2/UCP3. Though the topic is relevant and of potential interest for many researchers, the role of UCP2/UCP3 in MCU activity is highly debated. In addition, the provided data are not strong enough to support the main conclusion of the manuscript, the involvement of UCP2 in mitochondrial Ca²⁺ uptake. The manuscript can be improved conceptually by positively addressing the referee's concerns.

Major Comments:

1. According to the main hypothesis, MICU1 455Arg is the residue prone for methylation. This is supported mainly by showing methylation data of MICU197-444 and MICU197-476. However, the authors did not provide an experimental evidence to show MICU1455Arg mutant to be unmethylated.
2. It is not clear to reviewer the results of PRMT1 knock down and mitochondrial Ca²⁺ uptake. UCP2 interacts exclusively to methylated MICU1. In figure 2C through E, the authors performed the blockade of methylation by pharmacological blockers (AdOx/AMI) and genetic inhibition (siRNA PRMT1). So don't the authors expect non-methylated MICU1 that is unable to bind UCP2 and reduced mitochondrial Ca²⁺ uptake compared to control cells?
3. Do the authors think the methylation of MICU1455Arg to be independent of UCP2 abundance? As it can be seen from figure S1C, it is very clear that HUVEC and PAEC cells have reduced UCP2 levels. Overexpression of PRMT1 in HUVEC cells is supposed to decrease the mitochondrial calcium uptake (as per increased methylation of MICU1 and decreased UCP2 levels).
4. In figure 2g the authors look for the mRNA expression levels of core constituents of MCU complex including MCU, MCUB, MICU1, MCUR1, and EMRE. If the authors think these are core components of the MCU complex, why did the authors neglect MCUB and MCUR1 sequence analysis for identifying the potential arginine methylation sites?
5. The authors showed the interaction of methylated and unmethylated MICU1 with UCP2. Could

the authors show whether methylation of MICU1 have impact of interactome of MCU complex, particularly the interaction between MCU/MICU1 and EMRE/MICU1.

6. Does methylation of MICU1 regulates its interaction with UCP2/3? Authors should give the immunoprecipitation data for the MICU1-WT, MICU1Arg455Lys, and MICU1Arg455Phe mutant interaction with UCP2/3.

7. Please clarify, in Figure 3e, authors highlighted the band just lower than IgG heavy chain band as MICU1, but MICU1 molecular weight is ~56 kDa which is supposed to be higher than IgG heavy chain band. In contrast, panel C depicts the MCU band which lower than IgG indicating the right molecular weight (~39kDa). The reviewer feels that the MICU1 Western needs to be repeated.

8. Authors should give anti-Flag Ab western blot for the blots given in Figure 3c, d and e. Which would serve as an appropriate controls.

9. Page-3, line 3, authors stated that "expression of core constituents of the MCU complex" but they show expression for only MCU, MICU1, UCP2, UCP3, It is unclear to the reviewer that do authors claim these are the only core constituents of MCU complex? Clarify.

10. As authors showed the methylation of wt-MICU1 in Figure 3e, they should show MICU1Arg455Lys methylation status in PRMT1 overexpression and knockdown conditions.

11. In Supplementary Figure 4a, authors showed the MICU1 455 mutants distribution on the Mitochondria. They should show if these mutation have any effect on MICU1-MCU and MICU1-EMRE co-localization.

12. In Supplementary Figure 4a, mitochondrial network in 455F mutant looks altered as compare to WT and 455K mutant. Does this mutation have any significant effect on mitochondrial morphology?

13. In their discussion part the authors discussed "However, an increased PRMT1 activity that also results in methylation of MICU1 and, thus, a reduced mitochondrial Ca²⁺ uptake/activity would counteract the elevated demand of cancer cells on Ca²⁺-triggered ATP production in the mitochondria. By utilizing UCP2/3, cancer cells may be able to counteract the impact of an essential PRMT1 activity to mitochondrial Ca²⁺ uptake and, thus, re-establish activity of mitochondrial Ca²⁺ uptake to achieve Ca²⁺-triggered activation of ATP production". But as everyone knows that most of the cancer cells are glycolytic and not OXPHOS dependent (particularly the mitochondrial Ca²⁺ triggered activation of ATP production). Could the authors be more elaborative in this part?

Minor comments:

1. Instead of mentioning MICU1 as a regulator, the authors can claim it as a gatekeeper of MCU. The authors should cite the original articles but not the reviews.

2. There was a typo on PERMT1 in the summary paragraph, page 2 second paragraph (proofed), page 3 second paragraph (let)

3. Though MICU1 was identified by Vamsi and his group, the MICU1 as a gatekeeper for MCU activity was defined by Madesh and his group. Why the authors were biased in describing about MICU1 and completely ignoring the researcher's contribution?

4. Could the authors please take care of the typos throughout the manuscript?

5. Could the authors use the term knockdown or loss/reduction instead of diminution?

Reviewer #3 (Remarks to the Author):

PRMT-1mediated methylation of MICU1 determined the UCP2/3-dependency of mitochondrial Ca²⁺ uptake

What are the major claims of the paper?

The authors have investigated the controversial finding that UCP2/3 is involved in mitochondrial Ca²⁺ uptake. A macromolecular complex consisting of MCU, MICU1, and EMRE comprise the core of the Ca²⁺ pore-forming complex. They noted that a UCP2/3 effect was observed in some cell lines, but not others, and then used this difference to fish out the molecular details behind UCP2/3

function. Through a series of experiments, the authors wish to convey that Arg455 of MICU1 becomes arginine-methylated by PRMT1, which in turn affects how MICU1 oligomerizes, which in turn affects the sensitivity of MICU1 for Ca²⁺.

Are the claims novel? Will the paper be of interest to others in the field? Will the paper influence thinking in the field?

Yes.

This is a timely study and a wonderful tactical use of the different cell lines to investigate the controversial involvement of UCP2/3 in mitochondrial Ca²⁺ transport.

Are the claims convincing? If not, what further evidence is needed? Are there other experiments that would strengthen the paper further? How much would they improve it, and how difficult are they likely to be?

Are the claims appropriately discussed in the context of previous literature?

There are several aspects of the manuscript that need to be clarified/revisited in order to strengthen the data.

1. I know there is probably a requirement to be brief when it comes to methodology, but there are particular items that need to be present in the manuscript, not only to allow proper evaluation, but also for the scientific public to be able to repeat what has been discussed in this paper. The following comments are focused specifically on methods/strategy:

a) On page 3, the investigators begin to probe whether or not any of the pore-forming proteins are arginine methylated. They state that a "specific antibody against asymmetric protein arginine methylation" was used in the Western blots. In the Methods section, they do list Abcam as the source, but they need to identify which of the many Abs that Abcam sells is the one used in these studies. Additionally, although many antibodies are suggested to be specific for ADMA, many also recognize monomethylated arginine. Later in the manuscript (Figure 3 legend) they refer to an antibody that is specific for both MMA and ADMA... This has to be clarified. In order to show the specificity of this antibody for methylated MICU1, the investigators should perform a blocking experiment using a peptide methylated at position 455 (where they are using the Ab to assess MICU1 methylation).

b) The authors state that recombinant rat PRMT1 was used for in vitro methylations but they used a truncated version that is missing the first 10 residues. One, why? And two, the Nt of PRMT1 is thought to be involved in substrate selectivity (JBC article by Cote and co-workers), making the use of this truncated version a little complicated. In addition and even more importantly, the use of a "1:1 (vol) mixture of PRMT1: MICU1" for the in vitro reactions is totally ambiguous. The amount of each as well as the volume of the reaction should be noted. Additionally, an explanation of why the truncated form is used is warranted.

c) How did the investigators obtain/purify methylated MICU1 for their biophysical characterizations? I found no details regarding this. This is a critical detail considering the authors wish to convince the reader that methylated and nonmethylated MICU1 have different biophysical properties. In my opinion, none of the studies using "methylated MICU1" can be evaluated without this information.

d) Why do the authors think that a phenylalanine residue is a mimic for a methylated arginine? (page 5). Is there a published report showing such? I am unaware of any such report. This is key because the authors use this strategy to gain support for the functionality of the methylated arginine.

2. In addition to questions about methodology, there are places in the manuscript where the results do not necessarily support the conclusions that were stated, where additional experiments would strengthen the story, or where the data needs to be explained a little more:

a) Both the Hevel and Thompson groups have shown (using substrate profiling) that PRMT1 is very promiscuous and does not show consensus site recognition. It is unclear how the algorithms that the investigators used to identify potential methylation sites was trained or what the value of these

numbers mean.

b) On Page 3, the authors use AMI and Adox to inhibit PRMT1/PRMTs, respectively. In these experiments they look at the ability of the mitochondria to take up Ca^{2+} . In order to show that they have indeed worked as they suggest, the authors should blot the extracts from these cells with an anti-methylarginine antibody to show reduced methylation.

c) In Figures 2a and b, the authors show that the amount of ADMA[?]-methylated proteins is altered in some cell types. Although PRMT1 is the major isoform responsible for this modification, the experimental design does not rule out the participation of several other isoforms (2,3,4,6,and 8). This should be clarified. Also, by eye the values in the histogram look significant; please add statistical treatment to Fig 2a.

d) In Figures 3c, d, and e. pulldowns were performed to investigate which component of the pore-forming complex might be methylated. These blots have many additional bands (I gather both heavy and light chain Ab fragments represent at least 2 of the other bands?). Please clarify. Given the number of bands that show up in 3e when MICU1 is pulled down, how can you be sure which band you are looking at? Was this data normalized for the amount of protein pulled down? E.g., in 3e, was the same amount of MICU1 pulled down in each experiment? This would be an important control. When discussing the ramifications of the data in Fig 3, the conclusion that "These data approve MICU1 as a specific target of PRMT1"(top of page 5) appears too strong. The subsequent experiments help to support this conclusion later in the manuscript, but right here it seems premature.

e) In Figure 4 the investigators try to nail down if MICU1 is methylated and where. They also introduce two variants that represent a non-methylatable MICU1 (Lys substitution) and a methylation mimic (Phe). As noted above, I do not understand the logic behind the Phe substitution. Nonetheless, I will comment on the rest of the experiments. The truncation experiments are a good start but not a full-proof way to answer the question at hand. Truncating proteins can affect structure which may be a necessary component to PRMT recognition. There are several types of experiments that would be more convincing. For example, the investigators could look at radiolabel incorporation from 3H SAM into WT and their R455K mutant. Additionally, FLAG-MICU1 could be precipitated and the modification identified by mass spectrometry.

f) The authors investigate the dissociation constant (Kd) of MICU1 oligomerization using ITC. But the values they obtain (15.4 +/- 1.2uM versus 42.2 +/-28.0um) have some issues. These is a huge amount of error in the second figure, enough that it overlaps with the value of non-methylated MICU1. Additionally, as noted above, it is unclear how they even made methylated MICU1. Was recombinant PRMT1 still around when they did the experiments? Does in vitro methylation target the same arginine as in vivo? Or were more arginines methylated in vitro? Was the MICU1 stoichiometrically methylated? There is no data given to show that there is indeed a difference in the MICU1 in the experiments that were performed, so there is no way I can evaluate this data beyond what I said above.

3. Syntax, grammar, labels, etc:

a) when discussing increased/decreased amounts of macromolecules (such as RNA or protein), please be sure to delineate which one you are talking about (e.g., page 3, line 6 of the first full paragraph reads "All cells tested expressed several PRMT isoforms" which may lead a reader to believe that the authors tested for the presence of PRMT protein isoforms when in fact, an RTPCR experiment was done to look for RNAs. It will bring clarity to the manuscript.

b) several places where unconventional words are used (e.g., page 2, first sentence of the first full paragraph reads " UCP2/3 were proofed to be fundamental")

c) Figure 3: on all the Western blots, a label that reads "MDMA AB" is present. I am assuming this represents the use of the Abcam Ab against methylarginines. I would suggest using a consistent term throughout the manuscript and then using it on these figures. In Figure 1, the Western is labeled with "aDMA AB". Were these blots probed with different antibodies?

If the manuscript is unacceptable in its present form, does the study seem sufficiently promising that the authors should be encouraged to consider a resubmission in the future?

Yes

Response to the referees:

We thank all referees very much for their overall positive judgments and their excellent comments and valuable guideline that helped us very much to improve our work. With all our energy and technical possibilities we could successfully elaborate the major concerns. Please notice that we have also used alternate techniques not suggested by the referees in order to provide essential new insights related to the comments of the reviewers.

Reviewer #1 (Remarks to the Author):

I find this work to be extremely interesting and insightful, albeit not necessarily for the message the authors intended to convey. I do believe that the title "PRMT1-mediated methylation of MICU1 determines the UCP2/3-dependency of mitochondrial Ca²⁺ uptake" properly describes their findings for immortalized cells. The crucial finding in the view of the reviewer is that the same does not hold for freshly harvested, non-immortalized HUVEC cells. Reliance on mitochondrial functions of immortalized cells is dramatically different from that of primary cultures and cells in situ. In fact, this paper may offer a resolution for many of the controversies regarding this issue (role of UCPs in calcium transport).

We thank the referee very much for her/his very positive statement.

I would support the paper, if the above reasoning would be accepted by the authors. They should adjust the title to reflect the findings in immortalized cells. Of course, to strengthen the point I am raising here, they could do some more work on non-immortalized cells to confirm or disconfirm their current possible conclusions.

This is an excellent point and we thank the referee very much for this insightful comment. Following the referee's valuable advice we have further tested various cell lines and could not find a specific pattern that indicates that only immortalized cells exhibit UCP2/3-dependence of the mitochondrial Ca²⁺ uptake route. However, since introduction of PRMT1 activity in the freshly isolated, short term cultured HUVEC established an UCP2/3-dependent MCU activity, we believe that this phenomenon correlates with PRMT1 activity rather any other processes related to cell immortalization (see the table below).

Cell Type	Origin	tested	UCP2/3 ko	UCP2/3 oe
Ea.hy926 / C8	Human Endothelial Cell Hybrid	√	↓	↑
HUVEC/ECFC	Human Umbilical Vein Endothelial Cell	P1 √	↔, +PRMT1oe ↓	n.t.
PAEC	Porcine Aortic Endothelial Cell	√	P1: ↔	n.t.
HeLa (S3)	Human Cervical Cancer Cell	√	↓	↑
MCF-7	Human Breast Metastatic Adenocarcinoma Cell	√	↓	n.t.

SHSY5Y	Human Neuroblastoma Cell	√	↔	n.t.
INS-1	Rat Pancreatic β-cell	√	↓	n.t.

However, we completely agree with the referees comment as an increased PRMT1 activity is associated in many, if not most/all cancers, thus, immortalization would often introduce UCP2/3-dependence. Accordingly, we have addressed this important point in line 273-279.

Reviewer #2 (Remarks to the Author):

The manuscript entitled "PRMT1-mediated methylation of MICU1 determines the UCP2/3-dependency of mitochondrial Ca²⁺ uptake" by Sokolowski et al., is a follow up of the previous works by the same group. The authors have focused on showing evidence for the UCP2/3 dependency of mitochondrial Ca²⁺ uptake via the methylation of MICU1 by PRMT1. From the biochemical point of view the authors argue the dependency of mitochondrial Ca²⁺ uptake by UCP2/UCP3. Though the topic is relevant and of potential interest for many researchers, the role of UCP2/UCP3 in MCU activity is highly debated. In addition, the provided data are not strong enough to support the main conclusion of the manuscript, the involvement of UCP2 in mitochondrial Ca²⁺ uptake. The manuscript can be improved conceptually by positively addressing the referee's concerns.

Major Comments:

1. According to the main hypothesis, MICU1 455Arg is the residue prone for methylation. This is supported mainly by showing methylation data of MICU197-444 and MICU197-476. However, the authors did not provide an experimental evidence to show MICU1455Arg mutant to be unmethylated.

This is an important point. We have addressed this issue by testing the PRMT1-mediated methylation of both mutants. The results are now provided as **Fig.4a** (MICU1-K) and **Supplementary Fig. 4a** (MICU1-F) and have been mentioned in lines 155-157

2. It is not clear to reviewer the results of PRMT1 knock down and mitochondrial Ca²⁺ uptake. UCP2 interacts exclusively to methylated MICU1. In figure 2C through E, the authors performed the blockade of methylation by pharmacological blockers (AdOx/AMI) and genetic inhibition (siRNA PRMT1). So don't the authors expect non-methylated MICU1 that is unable to bind UCP2 and reduced mitochondrial Ca²⁺ uptake compared to control cells?

Under basal conditions, these cell type exhibits largely enhanced PRMT1 activity (**Fig. 2**) leading to a methylation of most, if not all MICU1. This results in the sensitivity of mitochondrial Ca²⁺ uptake to UCP2/3 knock-down. The remaining uptake might be a mixture of the remaining UCP2/3 resisting the siRNA treatment (**Supplementary Fig. 1a**), non-methylated MICU1 and, most likely, MCU activation by high Ca²⁺ hot spots (Giacomello, M. *et al. Mol Cell* **38**, 280–290 (2010)) that are sufficient high to activate the MCU. Upon the inhibition of PRMT1 activity either by the inhibitor or the siRNA against PRMT1, most of the MICU1 is non-methylated and, thus, no UCP2/3 is required to get full activation of MCU activity. This has been summarized in lines 253-259.

3. Do the authors think the methylation of MICU1455Arg to be independent of UCP2 abundance? As it can be seen from figure S1C, it is very clear that HUVEC and PAEC cells have reduced UCP2 levels. Overexpression of PRMT1 in HUVEC cells is supposed to decrease the mitochondrial calcium uptake (as per increased methylation of MICU1 and decreased UCP2 levels).

This is an important point. Our data indicate that PRMT1-mediated methylation of MICU1 occurs independently of the presence or absence of UCP2 (**Fig. 3e** vs. **Supplementary Fig. 4a**). We have recently shown that MICU1 rearrangement by intracellular Ca^{2+} release occurs in less than 10% of the MICU1 proteins (Waldeck-Weiermair, *Sci Rep* 5:15602 (2015)) which is in line with the hot spot findings by Pozzan's group (Giacomello, M. *et al. Mol Cell* **38**, 280–290 (2010)). We assume that in view of the relative small numbers of hot spots there will be a sufficient high number of UCP2/3 molecules present even in HUVEC cells to interact with methylated MICU1.

4. In figure 2g the authors look for the mRNA expression levels of core constituents of MCU complex including MCU, MCUB, MICU1, MCUR1, and EMRE. If the authors think these are core components of the MCU complex, why did the authors neglect MCUB and MCUR1 sequence analysis for identifying the potential arginine methylation sites?

Indeed, we cannot completely exclude the possibility of other proteins of the MCU complex to get methylated by PRMT1. Since we did not see methylation of MCU and the given homology between these MCUB, this protein was not tested upon PRMT1-mediated methylation. Hence, as in these particular cells, MCUR1 knock-down did not affect mitochondrial Ca^{2+} uptake in HeLa as well as Ea.hy926 cells in our experiments (please see figure below), MCUR1 was not included. Certainly that does not exclude these proteins and other, probably not discovered proteins of the MCU complex, to serve as PRMT1 targets and or scaffolds. Nevertheless such evaluation exceeds the frame of the present study and we hope our work will initiate such work in the close future. However, in view of our data with the MICU1 mutants we feel strong that the methylation of MICU1 is the key phenomenon of the engagement of UCP2/3. We have discussed this important point in lines 305 to 309.

Mitochondrial Ca^{2+} uptake upon addition of 100 μM histamine in the absence of extracellular Ca^{2+} in Ea.hy926 (left panel) and HeLa cells (right panel)

5. The authors showed the interaction of methylated and unmethylated MICU1 with UCP2. Could the authors show whether methylation of MICU1 have impact of interactome of MCU complex, particularly the interaction between MCU/MICU1 and EMRE/MICU1.

Indeed, this is a very interesting point. However, with all our highest respect, we feel that an evaluation of the whole interactome of the MCU complex upon methylation of MICU1 is due to the available tools an impossible task to be addressed within the given time of three months of revision. Our data show that methylation of MICU1 engages UCP2/3 and, thus, changes the interactome of the MCU complex. However, functionally we see that the essential dependence of MCU and EMRE remain independent of the methylation MICU1 and the engagement of UCP2/3.

6. Does methylation of MICU1 regulates its interaction with UCP2/3? Authors should give the immunoprecipitation data for the MICU1-WT, MICU1Arg455Lys, and MICU1Arg455Phe mutant interaction with UCP2/3.

We have addressed this important point by ITC as this technique allows us to visualize the interaction between methylated or non methylated MICU with UCP2/3 specifically. The new results can be find in new **Supplementary Fig. 5b**.

7. Please clarify, in Figure 3e, authors highlighted the band just lower than IgG heavy chain band as MICU1, but MICU1 molecular weight is ~56 kDa which is supposed to be higher than IgG heavy chain band. In contrast, panel C depicts the MCU band which lower than IgG indicating the right molecular weight (~39kDa). The reviewer feels that the MICU1 Western needs to be repeated.

We agree with the referee and have repeated this experiments using another, more specific antibody against asymmetrical dimethylation of arginines (aDMA) and normalized with the Flag antibody. This is now shown in the new **Fig. 3e**. For comparison with panels c & d, the previous blot is shown as **Supplementary Fig. 3a**. It is now obvious that heavy loading of the gel caused disturbances in the migration of the bands. The new blot (**Fig. 3e**) demonstrates clearly that MICU1 gets methylated by PRMT1.

8. Authors should give anti-Flag Ab western blot for the blots given in Figure 3c, d and e. Which would serve as an appropriate controls.

To address this point, the purity of the immunoprecipitation procedure was tested by comparing IgG control vs. control cell lysates. These blots are now provided as new **Supplementary Fig. 3b**.

9. Page-3, line 3, authors stated that "expression of core constituents of the MCU complex" but they show expression for only MCU, MICU1, UCP2, UCP3, It is unclear to the reviewer that do authors claim these are the only core constituents of MCU complex? Clarify.

According to the referee's comment we have now added the evaluation of MCUb, EMRE and MCUR1 to this graph shown as new **Supplementary Fig. 1c** and have rewritten this sentence according these results (lines 70-72).

10. As authors showed the methylation of wt-MICU1 in Figure 3e, they should show MICU1Arg455Lys methylation status in PRMT1 overexpression and knockdown conditions.

We thank the referee for raising up this important point and have conducted the respective experiments now shown as new **Fig 4a** (MICU1K) and Supplementary 4b (MICU1-F) and mentioned this point in lines 155-157.

11. In Supplementary Figure 4a, authors showed the MICU1 455 mutants distribution on the Mitochondria. They should show if these mutation have any effect on MICU1-MCU and MICU1-EMRE co-localization.

In line with the referee's suggestion, MCU-MICU1 and MCU-EMRE co-localization have been tested and added as new **Supplementary Fig. 4d** and were described in lines 161 to 162. As this technique was not included in the previous work, the respective technique has been explained in line 521 to 530.

12. In Supplementary Figure 4a, mitochondrial network in 455F mutant looks altered as compare to WT and 455K mutant. Does this mutation have any significant effect on mitochondrial morphology?

We thank the referee for this very insightful comment and have analyzed the impact of the MICU1 mutants on mitochondrial morphology. These data now given as new **Supplementary Fig. 4e** show that compared with wt MICU1 and the MICU1-K mutant, the methylation-mimicking mutant MICU1-F affects mitochondrial morphology by rounding up this organelle while the surface and volume remained similar. Since neither basal (**Supplementary Fig. 4c**) nor the amplitude or pattern of mitochondrial Ca^{2+} signals (**Figure. 4a, b**) were altered, one can exclude mitochondrial depolarization and/or changes of the pH as causes of these morphological changes by the MICU-F mutant. Hence, since neither the PRMT1 knock-down nor PRMT1 overexpression affected mitochondrial shape comparable with MICU1-F mutant (please see data below), we assume that these morphological changes might be specific for this given mutant and are not related to PRMT1 methylation. This data have been described in lines 162 to 169.

13. In their discussion part the authors discussed "However, an increased PRMT1 activity that also results in methylation of MICU1 and, thus, a reduced mitochondrial Ca^{2+} uptake/activity would counteract the elevated demand of cancer cells on Ca^{2+} -triggered ATP production in the mitochondria. By utilizing UCP2/3, cancer cells may be able to counteract the impact of an essential PRMT1 activity to mitochondrial Ca^{2+} uptake and, thus, re-establish activity of mitochondrial Ca^{2+} uptake to achieve Ca^{2+} -triggered activation of ATP production". But as everyone knows that most of the cancer cells are glycolytic and not OXPHOS dependent (particularly the mitochondrial Ca^{2+} triggered activation of ATP production). Could the authors be more elaborative in this part?

With all respect but in this particular point we have to disagree with the referee. Indeed, cancer cells have an upregulated glycolysis. However, due to a specific pyruvate kinase isoenzyme M2 enhanced glycolysis has a bottleneck at the conversion of phosphoenolpyruvate to pyruvate and the accumulation of the intermediate products of the glycolysis drives the production of components needed for cancer cell growth (nucleic acids, phospholipids, etc.) (e.g. see the work of Sybille Mazurek). Hence OXPHOS activity is, at least partially, fueled by amino acids and proteins and, in lesser extent fatty acids, entering the citrate cycle and fire mitochondrial oxidation. Because of the lack of pyruvate, malate accumulates and is converted to huge amounts of lactate, known as the Otto Warburg phenomenon (=enhanced glycolysis & high lactate levels). Recent work strikingly shows the mitochondrial engagement, particular the enhanced communications between mitochondria and the ER to be very important in cancer survival (please see e.g. Thomas N. Seyfried, Cancer as a mitochondrial metabolic disease. *Front Cell Dev. Biol.* **3**, 1-12 (2015), or very recently Cardenas et al. Selective Vulnerability of Cancer Cells by Inhibition of Ca^{2+} Transfer from Endoplasmic Reticulum to Mitochondria. *Cell Reports* **14**, 2313-2324 (2016)). Considering our present findings, we think its rather striking that most cancer cells have an enhanced PRMT1 activity. However, we completely agree with the referee's impression that there is lots of open question and we hope that this contribution will inspire to explore the role of posttranslational modifications of the mitochondrial Ca^{2+} uptake machineries in cancer and other human diseases.

Minor comments:

1. Instead of mentioning MICU1 as a regulator, the authors can claim it as a gatekeeper of MCU. The authors should cite the original articles but not the reviews.

We are sorry for this irritating writing and have corrected this and have mentioned the gatekeeping function of MICU1 including the appropriate references in the introduction on line 35-36.

2. There was a typo on PERMT1 in the summary paragraph, page 2 second paragraph (proofed), page 3 second paragraph (let)

corrected

3. Though MICU1 was identified by Vamsi and his group, the MICU1 as a gatekeeper for MCU activity was defined by Madesh and his group. Why the authors were biased in describing about MICU1 and completely ignoring the researcher's contribution?

We apologize if the referee felt us biased against the great work of the "Madesh laboratory". Actually, we can assure the editor that we highly appreciate the work of Madesh and his team and we are entirely free against any sort of bias against someone. In fact we have cited his great work in the former reference 14 (now 14) (Mallilankaraman, K. *et al.* MCUR1 is an essential component of mitochondrial Ca^{2+} uptake that regulates cellular metabolism. *Nat. Cell Biol.* **14**, 1336–1343 (2012)), 16 (now 16) (Hoffman, N. E. *et al.* SLC25A23 augments mitochondrial Ca^{2+} uptake, interacts with MCU, and induces oxidative stress-mediated cell death. *Mol Biol Cell* **25**, 936–947 (2014)), and 42 (now 44) (Hoffman, N. E. *et al.* MICU1 motifs define mitochondrial calcium uniporter binding and activity. *CellReports* **5**, 1576–1588 (2013)). However, we completely agree that we should also mention the work on MICU1 as gatekeeper that is now inserted as new reference 12 (Mallilankaraman, K. *et al.* MICU1 is an essential gatekeeper for MCU-mediated mitochondrial Ca^{2+} uptake that regulates cell survival. *Cell* **151**, 630-644) and the transcriptional modulation of MCU expression that is now included as new reference 61 (Shanmughapriya, S. Ca^{2+} signals regulate mitochondrial

metabolism by stimulating CREB-mediated expression of the mitochondrial Ca²⁺ uniporter gene MCU. *Sci. Signal* **8**, ra23 (2015)).

4. Could the authors please take care of the typos throughout the manuscript?

We apologize for the typos in the manuscript and have carefully proofread the manuscript to correct for typos.

5. Could the authors use the term knockdown or loss/reduction instead of diminution?

We have omitted the term “diminution” and used either “knock-down” or “reduction” instead.

Reviewer #3 (Remarks to the Author):

PRMT-1 mediated methylation of MICU1 determined the UCP2/3-dependency of mitochondrial Ca²⁺ uptake

What are the major claims of the paper?

The authors have investigated the controversial finding that UCP2/3 is involved in mitochondrial Ca²⁺ uptake. A macromolecular complex consisting of MCU, MICU1, and EMRE comprise the core of the Ca²⁺ pore-forming complex. They noted that a UCP2/3 effect was observed in some cell lines, but not others, and then used this difference to fish out the molecular details behind UCP2/3 function. Through a series of experiments, the authors wish to convey that Arg455 of MICU1 becomes arginine-methylated by PRMT1, which in turn affects how MICU1 oligomerizes, which in turn affects the sensitivity of MICU1 for Ca²⁺.

Are the claims novel? Will the paper be of interest to others in the field? Will the paper influence thinking in the field?

Yes.

This is a timely study and a wonderful tactical use of the different cell lines to investigate the controversial involvement of UCP2/3 in mitochondrial Ca²⁺ transport.

Thank you very much for your kind and encouraging words

Are the claims convincing? If not, what further evidence is needed? Are there other experiments that would strengthen the paper further? How much would they improve it, and how difficult are they likely to be?

Are the claims appropriately discussed in the context of previous literature?

There are several aspects of the manuscript that need to be clarified/revisited in order to strengthen the data.

1. I know there is probably a requirement to be brief when it comes to methodology, but there are particular items that need to be present in the manuscript, not only to allow proper evaluation, but also for the scientific public to be able to repeat what has been discussed in this paper. The following comments are focused specifically on methods/strategy:

a) On page 3, the investigators begin to probe whether or not any of the pore-forming proteins are arginine methylated. They state that a "specific antibody against asymmetric protein arginine methylation" was used in the Western blots. In the Methods section, they do list Abcam as the source, but they need to identify which of the many Abs that Abcam sells is the one used in these studies. Additionally, although many antibodies are suggested to be specific for ADMA, many also recognize monomethylated arginine. Later in the manuscript (Figure 3 legend) they refer to an antibody that is specific for both MMA and ADMA.... This has to be clarified. In order to show the specificity of this antibody for methylated MICU1, the investigators should perform a blocking experiment using a peptide methylated at position 455 (where they are using the Ab to assess MICU1 methylation).

Thank you for pointing to this issue: At the beginning of the study the MDMA antibody from Abcam was used for the Western blots showing the influence of PRMT1 on UCP2, MCU and MICU1. Since a clear difference in the methylation level of MICU1 could be detected upon overexpression or silencing PRMT1, this antibody appears suitable for revealing the effect of PRMT1 on these proteins. Nevertheless, to get a more specific information about the PRMT1

methylation status in experiments with whole cell lysates we used the asymmetric dimethyl arginine detecting antibody (aDMA) for all further experiments.

During the revision process the MICU1 Western blot was repeated (**new Fig. 3e**) using the (aDMA) antibody and the same result as with MDMA antibody was obtained. To make this point more clear, we added the necessary information in the legends as well as in the main text body. Accordingly we are presume that these experiments are sufficient to demonstrate the specific MICU1 methylation, since the same results were obtained with two different arginine methylation antibodies against MDMA and aDMA and, therefore, no blocking experiments were performed. (Methods lines 480-496).

b) The authors state that recombinant rat PRMT1 was used for in vitro methylations but they used a truncated version that is missing the first 10 residues. One, why? And two, the Nt of PRMT1 is thought to be involved in substrate selectivity (JBC article by Cote and co-workers), making the use of this truncated version a little complicated. In addition and even more importantly, the use of a "1:1 (vol) mixture of PRMT1: MICU1" for the in vitro reactions is totally ambiguous. The amount of each as well as the volume of the reaction should be noted. Additionally, an explanation of why the truncated form is used is warranted.

We thank the reviewer for her/his comments. We used a truncated version of rat PRMT1 lacking the first 10 amino acids for the *in vitro* arginine methylation experiments. This version is identical to human PRMT1v1 (older nomenclature; now Q99873-3) except a single amino acid exchange (H161 is Y in human). We have updated the methods section to include this information.

The vast majority of human and rat/mouse expressed sequence tags represent splicing version 1 (Q99873-3), which encodes a protein of 353 amino acids (Zhang et al., Structure. 2003 May;11(5):509-20.). The PRMT1 version we used is the rat PRMT1v1 (Q63009), which is identical to the mouse protein and differs from the human enzyme at only one position (H161 is Y in human). The previously reported human version 1 sequences (Q99873-3) lack the first ten amino acids of the rat sequence. Indeed, Cote and coworkers identified different activities of PRMT1 isoforms. However, all PRMT1 isozymes methylated an apparently similar set of cellular proteins, and differences in substrate specificity were observed mainly between PRMT1v1 (older naming; Q99873-3) and -v2 (older naming; Q99873-1). Given that the majority of rat expressed sequence tags represent splicing version 1 (Q99873-3), and that this version has previously been used successfully by many other labs including the Wahle lab, the Cheng lab etc. we are confident that the PRMT1 version used is well suited. We now provide more details for the experimental conditions used and have updated the methods section accordingly (lines 574-597).

c) How did the investigators obtain/purify methylated MICU1 for their biophysical characterizations? I found no details regarding this. This is a critical detail considering the authors wish to convince the reader that methylated and nonmethylated MICU1 have different biophysical properties. In my opinion, none of the studies using "methylated MICU1" can be evaluated without this information.

We wish to apologize for the confusion raised and have extended the description of protein expression and purification (lines 545-573), *in vitro* (lines 574-597) methylation and ITC (lines 598-605).

d) Why do the authors think that a phenylalanine residue is a mimic for a methylated arginine? (page 5). Is there a published report showing such? I am unaware of any such report. This is key because the authors use this strategy to gain support for the functionality of the methylated arginine.

Phenylalanine has been the residue has been proposed to mimic the constitutively arginine-methylated state (Mostaqul Huq, M. D., Gupta, P., Tsai, N. P., White, R., Parker, M. G., and Wei, L. N. (2006) Suppression of receptor interacting protein 140 repressive activity by protein arginine methylation. EMBO J. 25, 5094–5104) and has been used recently by several groups to mimic arginine methylation (DOI 10.1074/jbc.M111.289496; 10.1074/jbc.M113.491092). We have updated the manuscript to include this information (lines 154-157) and added a respective reference (Huq, M. D., Gupta, P., Tsai, N. P., White, R., Parker, M. G., and Wei, L. N. (2006) Suppression of receptor interacting protein 140 repressive activity by protein arginine methylation. EMBO J. 25, 5094–5104) (now reference 46).

2. In addition to questions about methodology, there are places in the manuscript where the results do not necessarily support the conclusions that were stated, where additional experiments would strengthen the story, or where the data needs to be explained a little more:

a) Both the Hevel and Thompson groups have shown (using substrate profiling) that PRMT1 is very promiscuous and does not show consensus site recognition. It is unclear how the algorithms that the investigators used to identify potential methylation sites was trained or what the value of these numbers mean.

We decided to use computational prediction of arginine methylation sites as starting point for our mutational strategies. The webserver which is based on the cited publication by Shao-Ping Shi et al. (<http://dx.doi.org/10.1371/journal.pone.0038772>) uses a computational approach (enhanced feature encoding scheme) trained on a set of data for extracting informative amino acids features. This included sparse property coding (SPC; divided the 20 amino acid residues into four different groups according to their hydrophobicity and charged character), normalized van der Waals volume (VDWV), position weight amino acid composition (PWAA; sequence position information of amino acid residues around the methylation sites and non-methylation sites) and solvent accessible surface area (ASA; methylation sites tend to be solvent accessible). SPC and VDWV were utilized to characterize protein sequence information and physicochemical properties of amino acids surrounding methylation sites. PWAA and ASA were applied to represent sequence-order information and structural characteristic around methylation sites, respectively. A support vector machine (SVM) approach was used in combination with cross-validation to evaluate the performance of the SVM. The value of these numbers ranges between 0 (low probability) and 1 (high probability).

According to the authors, the training data were extracted from UniProtKB/Swiss-Prot database and PhosphoSitePlus and contained 98 proteins covering 246 experimental methylarginine sites, and 137 proteins covering 367 experimental methyllysine sites.

b) On Page 3, the authors use AMI and Adox to inhibit PRMT1/PRMTs, respectively. In these experiments they look at the ability of the mitochondria to take up Ca²⁺. In order to show that they have indeed worked as they suggest, the authors should blot the extracts from these cells with an anti-methylarginine antibody to show reduced methylation.

We thank the referee for this insightful comment and performed the experiments suggested and shown now in **Supplementary Fig. 2b** (lines 93-97) that both AMI-1 and AdOX strongly reduce methylation. Since AMI-1 is a PRMT1 selective inhibitor its effect compared with AdOX was reduced.

c) In Figures 2a and b, the authors show that the amount of ADMA[?]-methylated proteins is altered in some cell types. Although PRMT1 is the major isoform responsible for this modification, the experimental design does not rule out the participation of several other

isoforms (2,3,4,6,and 8). This should be clarified. Also, by eye the values in the histogram look significant; please add statistical treatment to Fig 2a.

Thank you very much for this important point. According to the referees suggestions we have performed respective experiments testing the involvement of PRMT2,3,4 and 6 and found no effect of a specific knock-down on the UCP2/3 dependency of mitochondrial Ca²⁺ uptake (lines 102-104 & new **Supplementary Fig. 2b**). Moreover the reasons for the exclusion of PRMT5,7,8&9 were included in the discussion section (lines 262-268).

d) In Figures 3c, d, and e. pulldowns were performed to investigate which component of the pore-forming complex might be methylated. These blots have many additional bands (I gather both heavy and light chain Ab fragments represent at least 2 of the other bands?). Please clarify. Given the number of bands that show up in 3e when MICU1 is pulled down, how can you be sure which band you are looking at? Was this data normalized for the amount of protein pulled down? E.g., in 3e, was the same amount of MICU1 pulled down in each experiment? This would be an important control. When discussing the ramifications of the data in Fig 3, the conclusion that "These data approve MICU1 as a specific target of PRMT1"(top of page 5) appears too strong. The subsequent experiments help to support this conclusion later in the manuscript, but right here it seems premature.

We thank the referee to point to some issues in our Western blots and have carefully addressed this issue:

Additional bands: Two of the additional bands are the heavy and light chain of the IgG (please compare to IgG control on the blots). To exclude impurities of the immunoprecipitation procedure which may cause additional bands, we performed Western blots comparing control IgG versus control lysates (also undergone immunoprecipitation procedure). No unspecific bands could be detected in those Western blots (**Supplementary Figure 3b**), showing that the bands in the Western blots (**Figure 3c,d,e**) are the proteins of interest and other proteins co-immunoprecipitated along with them. To confirm our previous findings, we additionally used the "VeriBlot for IP secondary antibody" for the repeated aDMA MICU1 Western blot (**Fig. 3e**). This antibody is only detecting native IgG-bands (not the denatured heavy and light IgG chains that are also eluted along with the flag tagged target protein during the immunoprecipitation process).

Normalization: The bands (**Fig. 3c,d and Supplementary 3a**) were normalized to the content of the corresponding protein (MCU, UCP2 and MICU1). During the revision process we did additional MICU1 Western blots which were normalized by using the Flag-antibody (**Fig. 3e**).

According to the referees suggestions we rewrote this sentence (lines 148)

e) In Figure 4 the investigators try to nail down if MICU1 is methylated and where. They also introduce two variants that represent a non-methylatable MICU1 (Lys substitution) and a methylation mimic (Phe). As noted above, I do not understand the logic behind the Phe substitution. Nonetheless, I will comment on the rest of the experiments. The truncation experiments are a good start but not a full-proof way to answer the question at hand. Truncating proteins can affect structure which may be a necessary component to PRMT recognition. There are several types of experiments that would be more convincing. For example, the investigators could look at radiolabel incorporation from 3H SAM into WT and their R455K mutant. Additionally, FLAG-MICU1 could be precipitated and the modification identified by mass spectrometry.

We have included ITC data for the R455K mutant in the revised version of the manuscript (new **Supplementary Fig. 5b**). Summarizing we find that both, non-methylated and methylated MICU1 R455K show Ca²⁺ affinity comparable to the wild type protein whereas

Ca²⁺ binding is weakened in case of methylated wt MICU1. Furthermore, no formation of protein aggregates was observed for MICU1 R455K in the *in vitro* arginine methylation reaction that was a characteristic behavior of the wild-type protein.

f) The authors investigate the dissociation constant (Kd) of MICU1 oligomerization using ITC. But the values they obtain (15.4 +/- 1.2uM versus 42.2 +/-28.0um) have some issues. These is a huge amount of error in the second figure, enough that it overlaps with the value of non-methylated MICU1. Additionally, as noted above, it is unclear how they even made methylated MICU1. Was recombinant PRMT1 still around when they did the experiments? Does in vitro methylation target the same arginine as in vivo? Or were more arginines methylated in vitro? Was the MICU1 stoichiometrically methylated? There is no data given to show that there is indeed a difference in the MICU1 in the experiments that were performed, so there is no way I can evaluate this data beyond what I said above.

To address these points we have extended the methods section with further experimental details, added ITC data for the R455K mutant and carried out ITC measurements in replicates. PRMT1 was removed after *in vitro* arginine methylation using Ni-NTA affinity chromatography and SEC. Using these additional polishing steps and using replicate ITC measurements we found that methylation of MICU1 significantly reduced its sensitivity for Ca²⁺ from a dissociation constant of 5.4 ± 0.7 μM for non-methylated MICU1 WT to 14.0 ± 1.5 μM binding of Ca²⁺ to methylated MICU1. The higher affinity can be explained because we have added additional clean-up steps and SEC (please see Methods section, lines 545-573) to get rid of minor concentrated aggregates and impurities. ITC data of MICU1 R455K showed no significant change in Ca²⁺ binding before and after methylation reaction (6.8 ± 0.7 μM and 6.0 ± 0.9 μM, respectively), indicating that R455 is the only residue modified by PRMT1 (new **Supplementary Fig. 5b**).

3. Syntax, grammar, labels, etc:

a) when discussing increased/decreased amounts of macromolecules (such as RNA or protein), please be sure to delineate which one you are talking about (e.g., page 3, line 6 of the first full paragraph reads "All cells tested expressed several PRMT isoforms" which may lead a reader to believe that the authors tested for the presence of PRMT protein isoforms when in fact, an RTPCR experiment was done to look for RNAs. It will bring clarity to the manuscript.

Thank you, we have corrected our work accordingly (lines 80-81 and lines 70-72).

b) several places where unconventional words are used (e.g., page 2, first sentence of the first full paragraph reads "UCP2/3 were proofed to be fundamental")

We went through the paper and tried to erase unconventional (German) phrases

c) Figure 3: on all the Western blots, a label that reads "MDMA AB" is present. I am assuming this represents the use of the Abcam Ab against methylarginines. I would suggest using a consistent term throughout the manuscript and then using it on these figures. In Figure 1, the Western is labeled with "adMA AB". Were these blots probed with different antibodies?

Thank you for this suggestion. We have clarified this point throughout the whole paper by clearly mentioning the antibody used for the vary blot shown (please see also our response to comment 1a).

If the manuscript is unacceptable in its present form, does the study seem sufficiently promising that the authors should be encouraged to consider a resubmission in the future?
Yes

We hope that our huge efforts to address the points raised satisfy the referee.

Reviewers' comments:

Reviewer #1 (Remarks to the Author):

While I find the majority of the responses of the authors adequate, and, I continue to like the work, they refused to change the title of the paper to reflect the conditions they we're studying. I am not sure why that is, but it certainly concerns this reviewer as many of the issues the other reviewer raised revolve around the pitfalls the in vitro systems they studied present. Arrogance of the title would simply undermine the authors credibility in biology of uncoupling proteins.

Reviewer #2 (Remarks to the Author):

This reviewer is pleased now based on the authors response and new additional experiments.

Before publication, authors must carefully cite reference 7 pertinently.

The Vais/Foskett paper (Cell Reports, 2016, v14, p403) suggests that a matrix-localized C-terminal EMRE domain is responsible for a matrix Ca^{2+} -dependent inactivation phenomenon at relatively low matrix Ca^{2+} levels; further, they made mutations and truncation within EMRE C-terminal domain, showing that the Ca^{2+} -dependent inactivation of the current can be abolished. Considering the topology of EMRE remains controversial with several recent papers (Tomar et al., Cell Reports, 2016 [published online]; Tsai et al., 2016, eLife, v5; Yamamoto et al., 2016, Biochim Biophys Acta, v1857, p831) suggesting an opposite orientation to that suggested by Vais/Foskett, the molecular mechanisms behind the electrophysiological observations in this paper are unclear.

Reviewer #3 (Remarks to the Author):

The authors have addressed nearly all of the concerns from the previous submission.

Current concerns:

1. The use of antibodies and truncation mutants to identify post-translational modifications is a sketchy business. For example, several enzymes, including PRMT1, have been shown to require distal amino acid interactions in order to promote methylation (see Paul Thompson's work). What if Arg455 is not the site of methylation but is required as an interaction site in order to get sufficient binding for catalysis? Using the truncated protein would remove this residue, making it look like Arg455 was the site of methylation. Certainly PRMT1 seems to play a role in the overall process that the paper describes. However, I would be careful in attributing methylation specifically to residue 455 since you do not have any MS evidence and you do not show radiolabel incorporation. Furthermore, target sites identified in vitro are not always the same as those identified in vivo.
2. Lines 81-84 discuss an experiment that the authors say measures the PRMT1 activity in the cells. This experiment does not measure activity. It measures the methylation status of the cells which is a result of all PRMT activities. Furthermore, since we do not yet know of a demethylase, this experiment does not necessarily report on the current PRMT activities, only that which has occurred over the lifetime of the endogenous substrate proteins.
3. Line 75 subheading and text in this paragraph uses the phrase "PRMT1..engages..." I would suggest using another word since this gives in the impression that PRMT1 binds to something.
4. Lines 297-298. The authors state that the in silico approach predicted methylation at R455 as "the most promising site". This is totally untrue. Looking at Supp table 1, there are many sites with equal or greater probabilities that were predicted. This sentence should be removed.
5. I think Figure 3f is missing???

Response-to-the-referees:

We thank the referees very much for their fair and very helpful advices. We apologize, if we failed in our previous revision to address all points adequately and have done our very best to be more successful this time. Please find below a point-to-point reply to the individual points raised including the lines in the manuscript where we addressed the respective issue (the changes are also highlighted in red).

Thank you!

Reviewer #1 (Remarks to the Author):

While I find the majority of the responses of the authors adequate, and, I continue to like the work, they refused to change the title of the paper to reflect the conditions they we're studying. I am not sure why that is, but it certainly concerns this reviewer as many of the issues the other reviewer raised revolve around the pitfalls the in vitro systems they studied present. Arrogance of the title would simply undermine the authors credibility in biology of uncoupling proteins.

We apologize and feel very sorry that the referee had the impression that we acted arrogant. We can assure the referee that this was absolutely not our intension but we have to admit that we obviously missed the point. We do have evidence that the phenomenon we describe occurs independent of the type of cell status (i.e. non-immortalized vs. immortalized vs. cancer) but always upon PRMT1 activity. This becomes obvious when we used freshly isolated endothelial cells and could turn them to become UCP2/3 dependent in their mitochondrial calcium uptake by expression of PRMT1 (Fig. 2f). Even more strikingly, we recognized that when keeping freshly isolated piglet aortae endothelial cells in culture for more than 4 passages, they develop increased PRMT1 activity and, thus, became UCP2/3-dependent with respect to their mitochondrial calcium uptake. While this study is far for being completed, the results together with the results shown in Fig. 2f may indicate that the reported phenomenon occurs also in freshly isolated and cultured cells. **However, we completely agree with the referee that in all our experiments we are using cells in culture, thus, we thankfully follow the referee's advice and will change the title as suggested by the referee in: "PRMT1-mediated methylation of MICU1 determines the UCP2/3-dependency of mitochondrial Ca²⁺ uptake in immortalized cells"**

Reviewer #2 (Remarks to the Author):

This reviewer is pleased now based on the authors response and new additional experiments.

Before publication, authors must carefully cite reference 7 pertinently.

The Vais/Foskett paper (Cell Reports, 2016, v14, p403) suggests that a matrix-localized C-terminal EMRE domain is responsible for a matrix Ca²⁺-dependent inactivation phenomenon at relatively low matrix Ca²⁺ levels; further, they made mutations and truncation within EMRE C-terminal domain, showing that the Ca²⁺-dependent inactivation of the current can be abolished. Considering the topology of EMRE remains controversial with several recent papers (Tomar et al., Cell Reports, 2016 [published online]; Tsai et al., 2016, eLife, v5; Yamamoto et al., 2016, Biochim Biophys Acta, v1857, p831) suggesting an opposite orientation to that suggested by Vais/Foskett, the molecular mechanisms behind the electrophysiological observations in this paper are unclear.

We thank the referee very much for this insightful comment and have cited the paper of Vais et al. more properly and indicated on the existing controversy and also cited other manuscripts mentioned by the referee (new references 8-10) in line 30.

Reviewer #3 (Remarks to the Author):

*The authors have addressed nearly all of the concerns from the previous submission.
Current concerns:*

*1. The use of antibodies and **truncation mutants** to identify post-translational modifications is a sketchy business. For example, several enzymes, including PRMT1, have been shown to require distal amino acid interactions in order to promote methylation (see Paul Thompson's work). What if Arg455 is not the site of methylation but is required as an interaction site in order to get sufficient binding for catalysis? Using the truncated protein would remove this residue, making it look like Arg455 was the site of methylation. Certainly PRMT1 seems to play a role in the overall process that the paper describes. However, I would be careful in attributing methylation specifically to residue 455 since you do not have any MS evidence and you do not show radiolabel incorporation. Furthermore, target sites identified *in vitro* are not always the same as those identified *in vivo*.*

Thank you very much for this important comment. We completely agree that the ultimate and most desired proof would be mass spectrometric analysis of MICU1. In fact, we carried out extensive MS analysis of *in vitro* methylated MICU1 using enzymatic digests. However, even when using 4 different enzymes we could not completely cover the MICU1 sequence. In particular and much to our frustration we miss the peptides harboring R455. This might be due to the severe aggregation-prone properties of this region which is enhanced by arginine methylation and which could lead to precipitation of the corresponding peptides upon digestion. Accordingly we admit that due to these technical issues we could not include MS data. We have also intensively discussed the potential use of radiolabel incorporation. However, the comparison of radiolabel incorporation in the wt and R455K mutant protein would not address the issue raised by the referee as if there would be no labeling, it could be the consequence of the lack of PRMT1 binding, thus, not necessarily mean it's the arginine 455 that becomes methylated.

However, as suggested also by other referees we presented a detailed characterization of the R455K (non-methylatable) and R455F (methylation mimicking) single point mutants in addition to the C-terminal truncation mutant. The rationale for the single point mutations was in fact to avoid interference of the mutation with PRMT1 binding, thus, addressing the referee's comment. The R455K (lysine instead of arginine) mutant retains similar properties (charge, side chain length) compared to the wt protein, but cannot be methylated by PRMT1. We see indeed that arginine methylation is lost in this mutant (Fig. 4a, Supplementary Fig. 4a).

Hence, if one would now assume that mutation of R455 would affect PRMT1-mediated methylation of a secondary site, then the R455F (methylation mimicking) mutant should shift MICU1 in-cell activity towards that of the unmethylated state, as it shows very different properties (charge, side chain length) compared to arginine. However, we find in the requested in-cell experiments that the R455F mutant mimics arginine methylation and shows in-cell activity comparable to the methylated state (Fig. 4c). Thus, interference of the mutation with potential secondary methylation states seems unlikely. We have now mentioned the alternate possibility mentioned by the referee and discussed this important point in more detail in the discussion on lines 300 to 309.

2. Lines 81-84 discuss an experiment that the authors say measures the PRMT1 activity in the cells. This experiment does not measure activity. It measures the methylation status of the cells which is a result of all PRMT activities. Furthermore, since we do not yet know of a demethylase, this experiment does not necessarily report on the current PRMT activities, only that which has occurred over the lifetime of the endogenous substrate proteins.

We completely agree with the referee and have change this sentence according the suggestion of the referee (line 83).

3. Line 75 subheading and text in this paragraph uses the phrase "PRMT1..engages..." I would suggest using another word since this gives in the impression that PRMT1 binds to something.

We thank the referee for this insightful comment rephrased this headings (line 76).

4. Lines 297-298. The authors state that the in silico approach predicted methylation at R455 as "the most promising site". This is totally untrue. Looking at Supp table 1, there are many sites with equal or greater probabilities that were predicted. This sentence should be removed.

This is an important point, thank you. In fact, in the first round of analysis that was performed at the beginning of our work there were much less sites predicted. Now there is a new release of the software that reported more sites. However, our argument why this is "the most promising site" builds on the combination of high methylation probability and functional relevance of the site based on the previously published crystal structure (Figure 3f). According to the referee's suggestion, we have rephrased this sentence to make this clear (lines 297 to 299).

5. I think Figure 3f is missing???

We are very sorry for this and have added this missing figure in the current submission.

REVIEWERS' COMMENTS:

Reviewer #3 (Remarks to the Author):

Authors have addressed all concerns.